# Learning interpretable network dynamics via universal neural symbolic regression

Jiao Hu[1,2], Jiaxu Cui ®[1,2] ✉ & Bo Yang ®[1,2] ✉

Discovering governing equations of complex network dynamics is a fundamental challenge in contemporary science with rich data, which can uncover the hidden patterns and mechanisms of the formation and evolution of complex phenomena in various fields and assist in decision-making. In this work, we develop a universal computational tool that can automatically, efficiently, and accurately learn the symbolic patterns of changes in complex system states by combining the excellent fitting capability of deep learning with the equation inference ability of pre-trained symbolic regression. We perform extensive and intensive experimental verifications on more than ten representative scenarios from fields such as physics, biochemistry, ecology, and epidemiology. The results demonstrate the remarkable effectiveness and efficiency of our tool compared to state-of-the-art symbolic regression techniques for network dynamics. The application to real-world systems including global epidemic transmission and pedestrian movements has verified its practical applicability. We believe that our tool can serve as a universal solution to dispel the fog of hidden mechanisms of changes in complex phenomena, advance toward interpretability, and inspire further scientific discoveries.

From the *Book of Changes* in ancient China to the dialectical thinking in the West, there exists a common philosophical thought that *the only constant is change*. Undoubtedly, scientists have been striving to discover the laws of changes in complex phenomena, attempting to explain, forecast, and regulate all things[1], such as emergence[2], chaos[3], synchronization[4], and critical phenomena[5]. As a widely accepted model, the changing patterns of states in complex systems are generally governed by a set of nonlinear differential equations[6] as $\dot{X}(t) = f(X(t), A, t)$, where $X(t) \in \mathbb{R}^{N \times d}$ is the system states at time $t$, $N$ and $d$ are the number of system components (nodes) and the state dimension, respectively. $A$ represents the additional information beyond the system states, such as the topological interactions among system components. As shown in the above formula, the dynamic behaviors exhibited by complex systems are primarily contingent upon the intricate interdependence between their internal interactions $A$ and dynamics governing equations $f$[6,7]. This motivates the search for reliable methodologies to formulate dynamics models of these complex systems[8–10]. However, a remarkable challenge arises in this

pursuit. In theoretically complete physical systems, the laws of change are delineated by well-discovered foundational principles[11–14], such as the electromagnetic laws governing the microscale exchanges among propelled particles. For the majority of complex systems, $f$ is agnostic, and equivalent foundational rules remain incompletely elucidated, such as global epidemic outbreak[15], extreme climate anomalies[16], and extinction of biological populations[17]. Consequently, this vague development has limited the exploration of these complex fields.

Fortunately, with data acquisition becoming increasingly accessible, the emergence of data-driven technologies has assisted in increasing the frequency with which human experts discover patterns of change in complex systems[18–22]. They can provide domain experts with richer and more meaningful inspiration across various fields, thereby accelerating the process of scientific discovery in areas such as mathematics[23,24] and physics[25,26]. Although much excellent work has been developed to reconstruct the symbolic models for low-variate dynamics of complex systems[27], e.g., bivariate shear flow equation[28], trivariate metapopulation epidemic model[29], and up to 9-variate

[1]College of Computer Science and Technology, Jilin University, Changchun, China. [2]Key Laboratory of Symbolic Computation and Knowledge Engineering of Ministry of Education, Jilin University, Changchun, China. ✉e-mail: cjx@jlu.edu.cn; ybo@jlu.edu.cn

Newton's law of gravitation[12], inferring governing equations for high-variate network dynamics remains important and challenging. This is primarily because the number of nodes $N$ in network dynamics is usually large, seen as in the epidemic spreading where the number of transmission areas or individuals ranges from tens to billions[15]. Additionally $d$ can be multi-dimensional, resulting in too many free variables ($N \times d$) in the equations and topological interactions with exponential growth, thereby increasing the complexity of inferring symbolic models[17].

At present, several cutting-edge studies are attempting to deal with the discovery of governing equations from network dynamics[22,30]. Two-phase sparse symbolic regression (TPSINDy)[8] simply parameterizes $f$ as a learnable linear combination of pre-defined orthogonal or non-orthogonal elementary function terms. Although the efficiency of equation inference is high, the rationality of the pre-defined function terms directly impacts the results, making sufficient and accurate domain expertise essential[31]. Another set of methods involves using graph neural networks (GNNs) to parameterize $f$ overcomes excessive expert knowledge[32]. However, due to the use of genetic programming (GP) to parse neural networks into symbolic equations, it brings the high-cost evolutionary search efficiency issue[11,16]. Therefore, how to effectively balance expert knowledge and computational costs, while ensuring high computational efficiency, introducing only a small amount or no expert knowledge, lowering the threshold for use, and efficiently discovering governing equations remains an open challenge.

In this work, to address the aforementioned challenges, we have developed a universal neural symbolic regression tool that can automatically, efficiently, and accurately learn the changing patterns of complex system states by combining the excellent fitting ability from deep learning and the equation inference capability from pre-trained symbolic regression. Our analysis of various complex network dynamics scenarios across fields such as physics, biochemistry, ecology, and epidemiology demonstrates that our tool exhibits remarkable effectiveness and efficiency. It can accurately and efficiently discover the governing equations of network dynamics, even when faced with noisy or incomplete topological data. It has also achieved excellent results in chaotic systems and real-world applications, including global epidemic transmission and pedestrian movements. We believe that our tool can serve as a new and general solution to eliminate the fog of hidden mechanisms of changes in complex phenomena from broad fields.

## Results

In this work, we wonder to develop a computational tool LLC (Learning Law of Changes) to discover the ordinary differential equations (ODEs) from observed data of network dynamics, i.e., $LLC : \mathcal{O} \to \mathcal{F}$, where observations $O = \{(\boldsymbol{X}(t), A, M_x, M_a)\}_{t=0}^{T} \subseteq \mathcal{O}$ and the ODEs $\boldsymbol{f} \in \mathcal{F}$. In fact, observations are sampled from continuous state data, which is obtained by solving the initial value problem with given initial states $\boldsymbol{X}(0)$ and topology $A$, i.e., $\boldsymbol{X}(t) = \boldsymbol{X}(0) + \int_0^t \boldsymbol{f}(\boldsymbol{X}(\tau), A, \tau)d\tau$. Note that $M_x$ and $M_a$ in the observations $O$ are the masks for observed states and topological structure, respectively, with the same shapes of $\boldsymbol{X}(t)$ and $A$, depicting the incompleteness of the observations, please see Supplementary Section I.D for detailed settings of specific scenarios. By configuring different masks, it can flexibly handle complex scenarios, such as heterogeneous network dynamics with local observations. The following section outlines the overall process of our tool and presents an analysis of experimental results across a range of network dynamics scenarios.

### The overall process of the LLC

When developing the computational tool, we encounter two stubborn issues: high-dimensional free variables and efficiency issues in symbolic inference. As mentioned earlier, due to the excessive number of

nodes and state dimensions, the search space of the symbolic models is too vast to identify the well-fitted dynamics equations for observed data[33]. Meanwhile, if we were to simply test all symbolic models by increasing equation length, it may take longer than the age of the universe until we get the target[12]. Even if de novo optimization algorithms such as evolutionary computation[34], reinforcement learning[35], and Monte Carlo tree search[36] are applied to find the objective symbolic model, it would still take an intolerable amount of time. More importantly, the performance of existing symbolic regression methods declines sharply when the number of variables exceeds three[37]. Therefore, to address both challenges, we employ the divide-and-conquer approach by introducing a few physical priors and using the powerful fitting capabilities from neural networks to achieve dimensionality reduction of high-dimensional network dynamic signals. We then use a pre-trained symbolic regression model to accelerate the efficiency of inferring dynamics equations.

**Decoupling network dynamics signals through neural networks and physical priors.** To alleviate the curse of dimensionality in network dynamics, we introduce a physical prior that the change of network states can be influenced by the system's own states and the states of its neighbors[6,8,38,39]. That is, we can decompose the governing equation $\boldsymbol{f}$ into two coupled components: self dynamics ($\boldsymbol{Q}^{(self)}$) and interaction dynamics ($\boldsymbol{Q}^{(inter)}$). Thus, the governing equations of network dynamics can be reformulated in node-wise form as $\dot{X}_i(t) = \boldsymbol{Q}_i^{(self)}(X_i(t)) + \sum_{j=1}^{N} A_{i,j} \boldsymbol{Q}_{i,j}^{(inter)}(X_i(t), X_j(t))$, where the subscripts $i$ and $j$ represent the corresponding nodes, $A_{i,j}$ is the adjacency matrix, $\boldsymbol{Q}_i^{(self)}$ and $\boldsymbol{Q}_{i,j}^{(inter)}$ capture the evolution process of their own states and the dynamical mechanism governing the pairwise interactions of their neighbors, respectively. At this moment, $\boldsymbol{f}$ is composed of a set of functions, i.e., $\boldsymbol{f} =: \{\boldsymbol{Q}_i^{(self)}, \boldsymbol{Q}_{i,j}^{(inter)}\}_{i,j=1}^{N}$. Actually, with the appropriate choice of $\boldsymbol{Q}_i^{(self)}$ and $\boldsymbol{Q}_{i,j}^{(inter)}$, the above equation can describe broad network dynamics scenarios[6,40]. Unlike the popular message-passing-based GNNs[41], the fusion with this physical prior has been empirically validated to effectively learn underlying dynamics and is more suitable for handling network dynamics scenarios[38]. Significantly, such formulation can achieve the desired dimensionality reduction for high-dimensional network dynamics, by learning the $d$-variate $\boldsymbol{Q}_i^{(self)}$ and $2d$-variate $\boldsymbol{Q}_{i,j}^{(inter)}$ instead of directly inferring the ($N \times d$)-variate $\boldsymbol{f}$. Note that if the network dynamics are homogeneous, meaning that the governing equations for all nodes are consistent, then only two functions need to be learned. However, if the dynamics are heterogeneous, multiple sets of functions need to be learned based on different configurations. For ease of description, we elaborate on homogeneous scenarios here, i.e., omitting subscripts $i$ and $j$, and the setting for heterogeneous scenarios can be acquired from subsequent experiments. As we known, the differential signal of network dynamics ($\dot{X}_i(t)$) can be calculated by $\boldsymbol{Q}^{(self)}$ and $\boldsymbol{Q}^{(inter)}$. Thanks to the excellent nonlinear fitting power from neural networks, we parameterize them separately using two neural networks, i.e., $\hat{\boldsymbol{Q}}_{\boldsymbol{\theta}_1}^{(self)}$ and $\hat{\boldsymbol{Q}}_{\boldsymbol{\theta}_2}^{(inter)}$. More specific architectures can be found in Methods. By fitting $\dot{X}_i(t)$, the parameters $\boldsymbol{\theta}_1$ and $\boldsymbol{\theta}_2$ of the neural networks can be learned through backpropagation. At this stage, we achieve decoupling of the dynamics signals by obtaining well-trained self dynamics and interaction dynamics functions.

**Parsing governing equations via pre-trained symbolic regression.** To open the black-boxes of the trained $\hat{\boldsymbol{Q}}_{\boldsymbol{\theta}_1}^{(self)}$ and $\hat{\boldsymbol{Q}}_{\boldsymbol{\theta}_2}^{(inter)}$, we employ symbolic regression techniques[27] to find the interpretable white-box equations that approximate the neural networks best from a vast equation space. However, the principle behind conventional symbolic

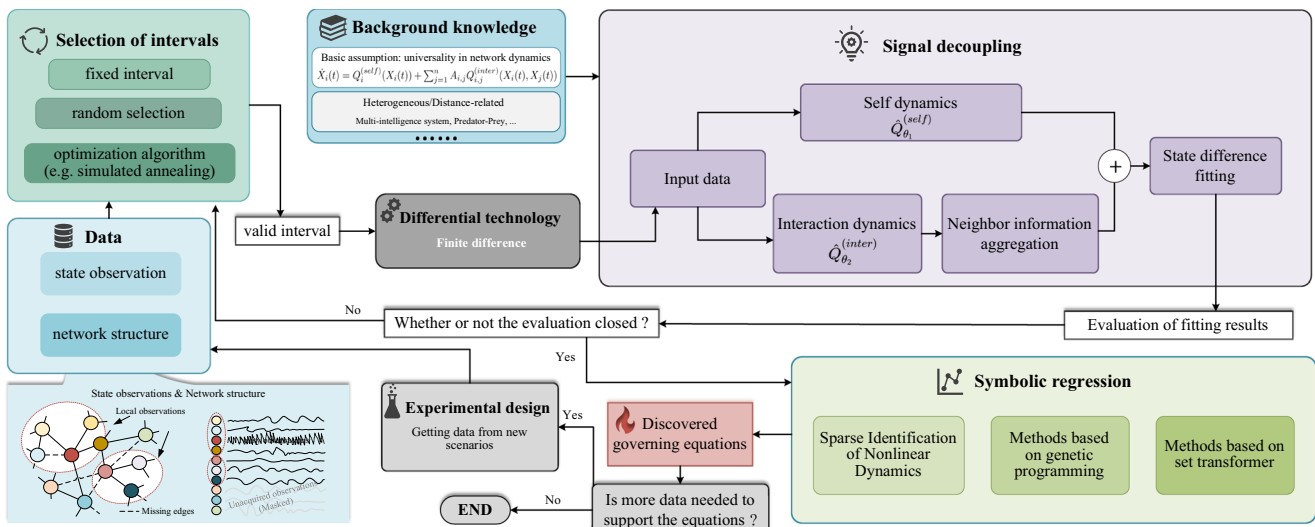

**Fig. 1 | The overall process of the LLC (Learning Law of Changes).** Observed data can be acquired from the initial experiments on the new scenario, including system states over time and topology, i.e. $O$. An interval selection strategy is to choose valid interval data and then we can get differentials, i.e. $\dot{X}_i(t)$, through finite difference on $X_i(t)$. By incorporating physical priors, the neural networks, i.e., $\hat{\boldsymbol{Q}}_{\boldsymbol{\theta}_1}^{(self)}$ and $\hat{\boldsymbol{Q}}_{\boldsymbol{\theta}_2}^{(inter)}$, are used to decouple the network dynamics signals and achieve variable reduction,

continuing until the fitting requirements are met. Otherwise, the training is repeated. After obtaining well-fitted neural networks, we use symbolic regression techniques to efficiently parse their approximate white-box equations. Of course, if additional observed data are needed to support the discovery, the experimental design can be revisited until the satisfactory governing equations of network dynamics are obtained, breaking the loop.

search algorithms is to seek optimal symbolic models from scratch when facing new tasks, often leading to lengthy execution times[34–36,42]. As a new generation of symbolic regression techniques, pre-trained transformer models[37] can improve the efficiency of equation inference. Such models are pre-trained on extensive sample data consisting of input-output points pairs from neural networks and equation[43], integrating massive equation knowledge. This enables them to rapidly derive equations from input-output points when addressing new tasks. When inferring a new equation, only a single forward propagation is required, significantly reducing the inference time, and then the constants in the equations can be optionally fine-tuned by using Broyden–Fletcher–Goldfarb–Shanno (BFGS)[44] or stochastic gradient optimization[45] to ensure inference accuracy. Specifically, we empirically use the NSRA[37] as our symbolic regression model, which is pre-trained on a dataset containing hundreds of millions of equations and has been shown to perform the best among several pre-trained models, to efficiently parse the symbolic equations of $\hat{\boldsymbol{Q}}_{\boldsymbol{\theta}_1}^{(self)}$ and $\hat{\boldsymbol{Q}}_{\boldsymbol{\theta}_2}^{(inter)}$. To our knowledge, this is the first attempt to introduce pre-trained models to discover governing equations of network dynamics.

**Assembling all modules to form an executable pipeline.** After overcoming the two core issues mentioned above, we built a tool to open up the path for discovering governing equations of network dynamics, as shown in Fig. 1. We obtain observed data from the initial experiments on the new scenario, including system states over time and topology (denoted as $O$), and use an interval selection strategy to choose valid interval data for getting differentials ($\dot{X}_i(t)$), through finite difference on $X_i(t)$. Then, combined with physical priors, the neural networks ($\hat{\boldsymbol{Q}}_{\boldsymbol{\theta}_1}^{(self)}$ and $\hat{\boldsymbol{Q}}_{\boldsymbol{\theta}_2}^{(inter)}$) decouple the network dynamics signals and achieve reduction in free variables, continuing until the fitting requirements are met. Otherwise, the training is repeated. Although we can directly train the neural networks via the adjoint method of neural ODE[25,32,38] based on observed data ($X_i(t)$), empirical results show that fitting the neural networks on differentials can bypass the high computational complexity associated with numerical integration in neural ODE and have more stable performance. After having well-fitted

$\hat{\boldsymbol{Q}}_{\boldsymbol{\theta}_1}^{(self)}$ and $\hat{\boldsymbol{Q}}_{\boldsymbol{\theta}_2}^{(inter)}$, we apply symbolic regression techniques to parse their approximate white-box equations efficiently. Of course, if additional observed data is required to support the discovery, the experimental design can be revisited until the satisfactory governing equations of network dynamics are acquired, thereby breaking out of the loop. Pseudo-code of the LLC describing the entire process can be found in Supplementary Section XI. The details of the modules in the pipeline will be provided in the Methods. Next, we will present and analyze the results of testing the LLC on a variety of network dynamics scenarios from different fields.

## Inferring one-dimensional network dynamics

To comprehensively assess the effectiveness of the LLC, we test it on six representative one-dimensional homogeneous network dynamics models, including Biochemical (Bio)[46], Gene regulatory (Gene)[47], Mutualistic Interaction (MI)[17], Lotka–Volterra (LV)[48], Neural (Neur)[49], and Epidemic (Epi)[50], which have widespread applications across various fields, including biology, ecology, epidemiology, genomics, and neuroscience, exhibiting diverse characteristics. All network dynamics are simulated on an Erdōs–Rényi (ER) network[51], where node degrees are drawn from a Poisson distribution with an average degree of $k = (N − 1)p$ and $p$ is the probability of edge creation, so as to produce the continuous system states, i.e., $\boldsymbol{X}(\cdot)$. We then apply the LLC to reconstruct governing equations from the observations $O = \{(\boldsymbol{X}(t), A, M_x, M_a)\}_{t=0}^{T}$ sampled from $\boldsymbol{X}(\cdot)$. Please refer to Supplementary Section I for a detailed description of the network dynamics models and their specific settings.

We compare against the state-of-the-art for interpretable dynamics learning in complex networks, namely, TPSINDy[8] and GNN+GP[11,38]. Due to the need to manually set function terms for the TPSINDy, we have set up various libraries with prior assumptions to obtain more comprehensive testing. For the GNN+GP, we use an empirically suitable GNN version for simulating network dynamics[38], and a high-performance GP-based symbolic regression tool, PySR[52], to derive equations. Also, we feed the output of our LLC as the discovered function terms into the TPSINDy, named as LLC+TPSINDy, to overcome its limitation of requiring a pre-defined library and demonstrate

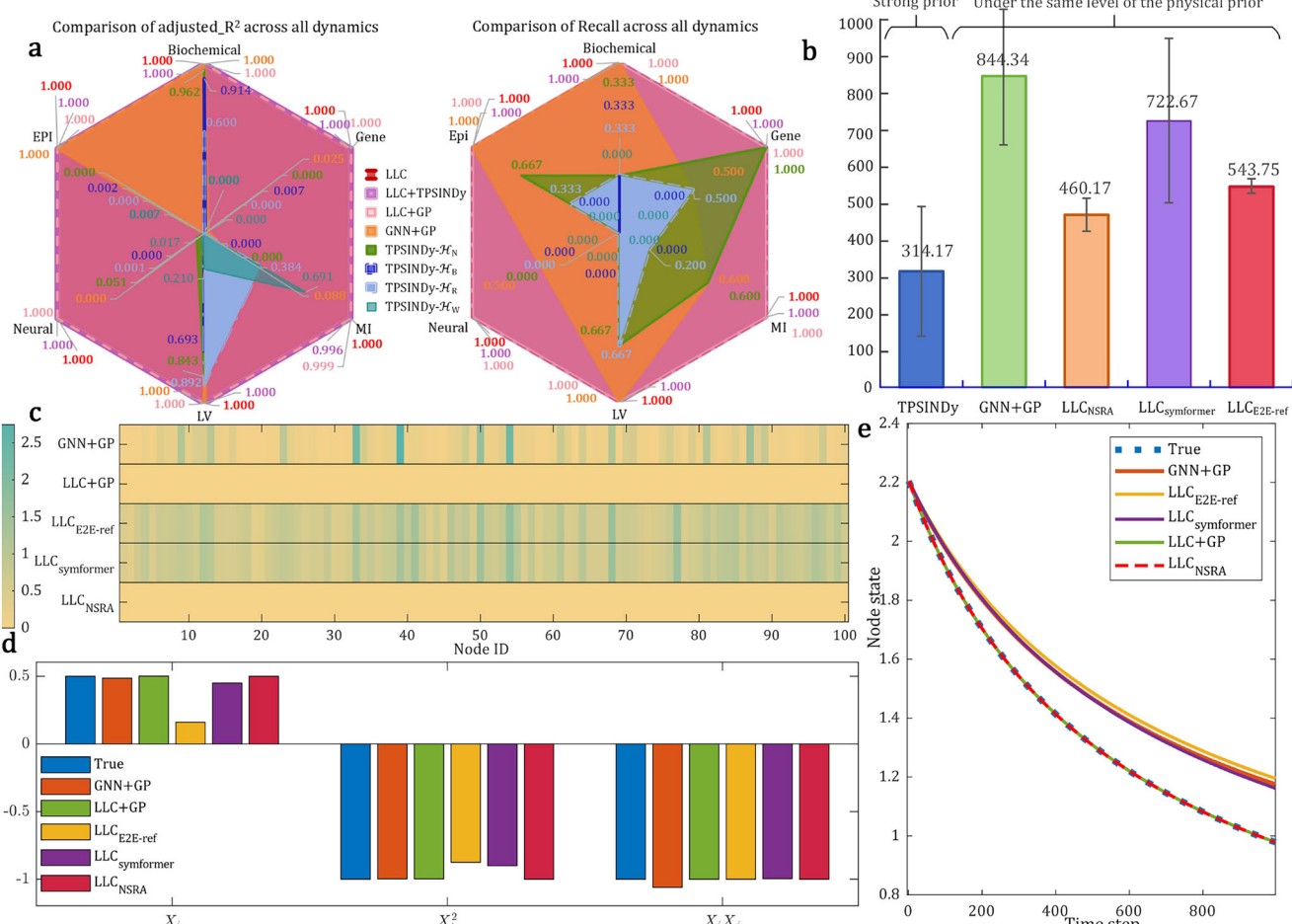

**Fig. 2 | Results of inferring one-dimensional homogeneous network dynamics.**
**a** Comparison of the accuracy on predictions (adjusted $R^2$ score) and discovered equations (Recall) for reconstructing dynamics from six scenarios, including Biochemical (Bio), Gene regulatory (Gene), Mutualistic Interaction (MI), Lotka-Volterra (LV), Neural (Neur), and Epidemic (Epi) dynamics. TPSINDy's results are highly dominated by its choice of function terms while our LLC significantly outperforms the comparative methods across all network dynamics scenarios. **b** Comparison of the average execution time across all dynamics for various methods. Note that the TPSINDy requires strong priors, i.e., decomposability of self and interaction dynamics, as well as pre-defined orthogonal elementary function terms. In contrast, others are based on the same level of assumption that only requires decomposability. By combining with transformer-based pre-trained symbolic regression in our pipeline, we achieve a good balance between efficiency and accuracy. **c** The NED (Normalized Estimation Error) between the predictive results produced by the discovered governing equations and ground truth in the LV scenario. **d** Comparison of the fitting coefficients in governing equations discovered by various methods. **e** Comparison of state prediction curves for an individual node.

our ability to discover new knowledge, i.e., effective function terms. Figure 2a shows the comparison of the accuracy on predictions (adjusted $R^2$ score) and discovered equations (Recall) for reconstructing dynamics from six scenarios. We observe that the choice of function terms significantly impacts the results of TPSINDy. The TPSINDy with a library that has the same function terms as ground truth (TPSINDy-$\mathcal{H}_N$ is a library that precisely contains the function terms of the objective equation) outperforms those with the libraries of containing only basic operations (TPSINDy-$\mathcal{H}_B$ only contains basic operations, such as polynomials, trigonometry, fractions, and exponential functions), lacking ground truth terms (TPSINDy-$\mathcal{H}_W$ lacks the function terms of the objective equation), and containing redundancy terms (TPSINDy-$\mathcal{H}_R$ includes not only $\mathcal{H}_N$ but also $\mathcal{H}_B$). Compared to the TPSINDy-based methodologies, the GNN+GP approach achieves high-precision state predictions and equation reconstructions in more scenarios, such as LV, Epi, and Bio. However, it struggles to deliver good results in more complex scenarios, like Gene, Neural, and MI. In contrast, our LLC-based achieves optimal performance across all network dynamics scenarios and significantly outperforms the

comparative methods on all metrics. Additional comparison results for other metrics, including MRE (Mean Relative Error), MAE (Mean Absolute Error), L2 error, as well as Precision, and the discovered equations for each scenario can be found in Supplementary Section IV. Note that the LLC+TPSINDy exhibits comparable performance and significantly improves upon TPSINDy, demonstrating that our LLC has discovered effective function terms to enhance existing methods. To further validate the effectiveness of the first part of our LLC, we compared against a variant that combines the first part of LLC and PySR, referred to as LLC+GP. Our findings indicate that LLC+GP successfully recovers the dynamic equation skeleton across all scenarios and outperforms the GNN+GP. This demonstrates that validating LLC's ability to provide effective decoupling supports the discovery of network dynamics. Consequently, it suggests that the proposed decoupling module, which incorporates physical priors, is more suitable for learning network dynamics than general GNNs. We also report the average time required for different methods to regress the expected equations in all scenarios, as shown in Fig. 2b. Undoubtedly, GNN+GP requires substantial computational time, i.e., ~12.9 min, to carefully

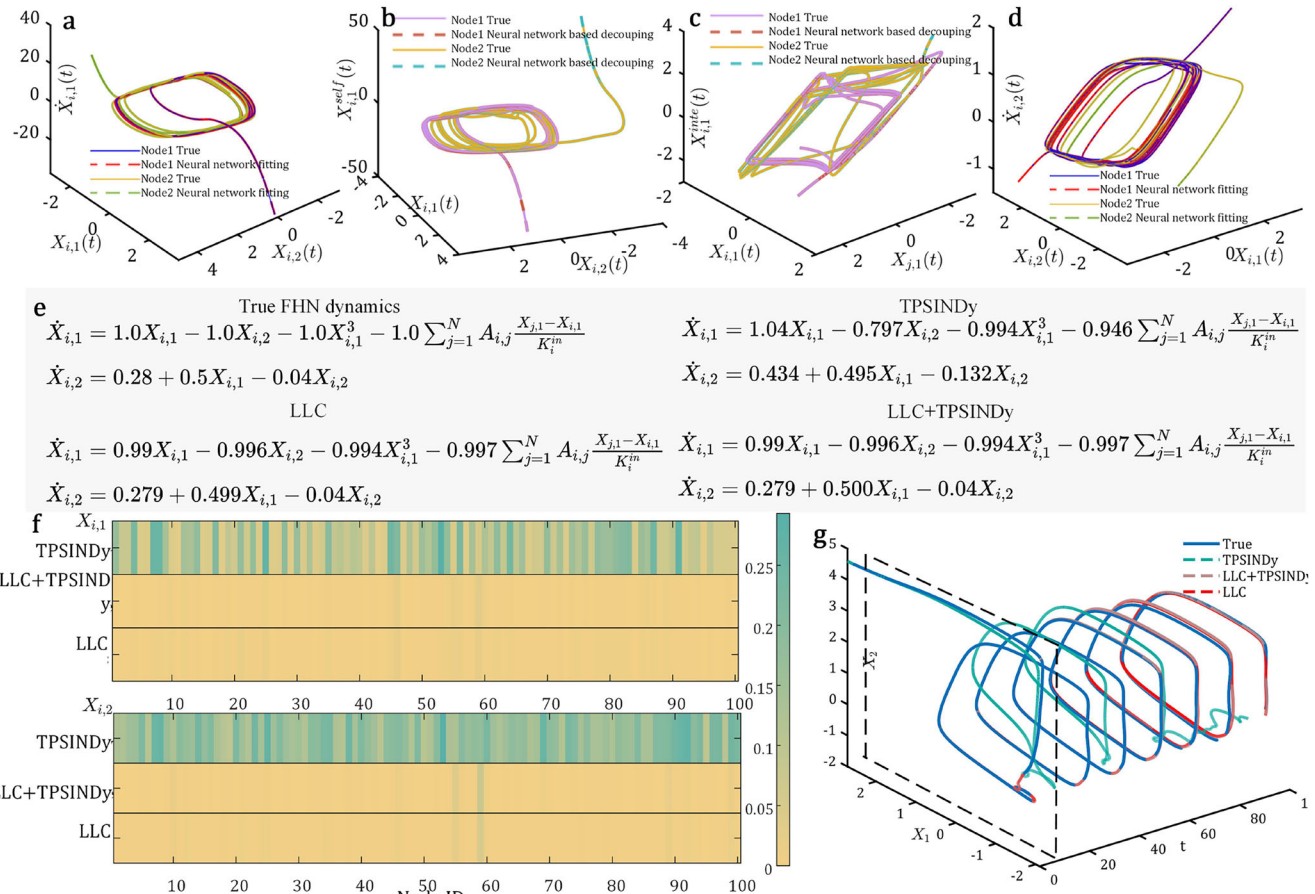

**Fig. 3 | Results of inferring the FitzHugh−Nagumo (FHN) dynamics. a** The fitting results of the first dimension ($\dot{X}_{i,1}(t)$) for a node by neural networks. **b** The decoupling results of the self dynamics for the first dimension for a node ($\hat{Q}_{\theta_1}^{(self)}$). **c** The decoupling results of the interaction dynamics for the first dimension for a node ($\hat{Q}_{\theta_2}^{(inter)}$). **d** The fitting results of the first dimension ($\dot{X}_{i,1}(t)$) for a node by neural networks. **e** Comparison of governing equations inferred by various methods. **f** Comparison of the normalized Euclidean distance (NED) between two trajectories, one generated from the inferred equations and the other from the true equations, with the horizontal axis representing the node index. **g** Comparison of the trajectories generated by the inferred and true equations on a Barabási−Albert network.

assemble and search expression trees. The introduction of more prior knowledge and the execution of simple linear regression have brought high efficiency to the TPSINDy, which takes only about 5.2 min, reducing time costs by 62.6%. Although the execution time of methods other than TPSINDy includes approximately 2.1 min of training computation required for decoupling signals, it is reasonable that the methods of using transformer-based pre-trained models as symbolic regression, including symformer[53], E2E-ref[54], and NSRA[37], have relatively low time costs, as they only perform a single, efficient forward propagation and optional constant optimization. Given only a small amount of expert knowledge, that is, the decomposability of self and interaction dynamics, LLC$_{NSRA}$ not only demonstrates high efficiency in regression (~6.5 min) but also has high accuracy, where the effectiveness of the discovery equations for each method is shown in Fig. 2c−e This is why we chose the NSRA as our symbolic regression module in the proposed pipeline, achieving a good balance between efficiency and accuracy.

Moreover, we introduce three types of noise, including zero-mean Gaussian noise, Poisson noise, and phase noise, into node states to evaluate the robustness of our tool, where the amount of noise is measured by the signal-to-noise ratio (SNR)[55]. As the SNR drops from 70 to 40 dB, ours always maintains an MSE close to 0 across all types of noise, while more noises weaken performance, especially in the case involving phase noise, but still significantly outperforms the TPSINDy (Supplementary Fig. 4), which possibly is a benefit brought by the

fitting of the neural network in signal decomposition and the variance minimization in loss function during training. For more details, please refer to Supplementary Section V.A. In addition, we randomly delete edges on the network to simulate situations where the topology is difficult to obtain fully. Supplementary Fig. 5 shows that the LLC can tolerate more missing edges. The main reason is that by decoupling network dynamics, the missing edges do not affect the learning for one's self dynamics, and only affect the amount of data available for learning interaction dynamics. The optional constant optimization in symbolic regression partially compensates for this.

### Inferring multi-dimensional and heterogeneous network dynamics
We apply the LLC to two multi-dimensional systems including FitzHugh−Nagumo (FHN) dynamics[56] and predator-prey (PP) system[57]. The FHN is a two-dimensional neuronal dynamics that brain activities governed by the FHN model, capturing the firing behaviors of neurons with two components: membrane potential ($X_{i,1}(t)$) and recovery variable ($X_{i,2}(t)$), which exhibits a quasi-periodic characteristic. A Barabási−Albert (BA) network is used to simulate the interactions among brain functional areas[58]. We get the observations of the sampling nodes by setting the masks, i.e., $M_x$ and $M_a$. As shown in Fig. 3a−d, our LLC can effectively decompose the self dynamics and interaction dynamics by training neural networks $\hat{Q}_{\theta_1}^{(self)}$ and $\hat{Q}_{\theta_2}^{(inter)}$. Moreover,

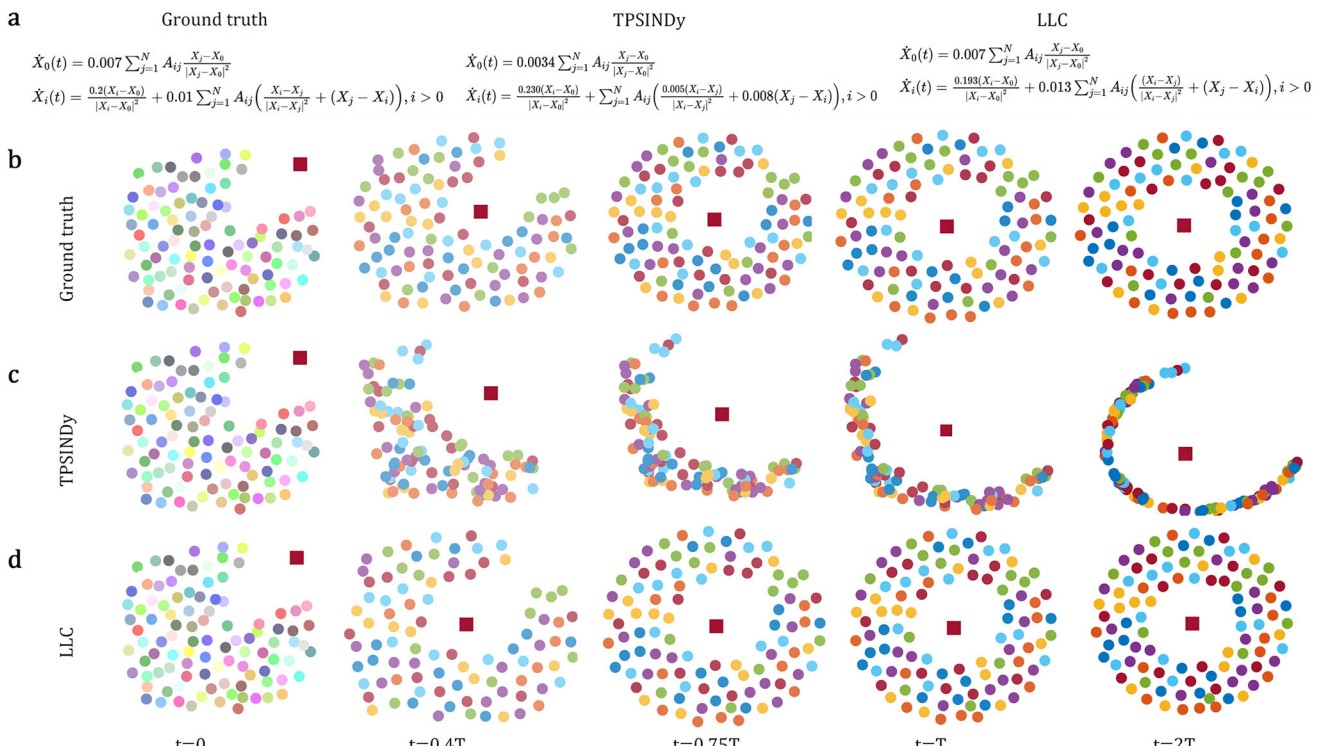

**Fig. 4 | Results of inferring the predator–prey (PP) system. a** Comparison of governing equations inferred by various methods. **b** The ground truth positions of a predator (square) and prey swarm (dots) over time. **c** The predictive positions generated by the governing equation inferred by the TPSINDy. **d** The predictive positions generated by the governing equation inferred by the LCC.

compared to the TPSINDy, which incorporates strong prior knowledge, specifically including polynomial functions $X_i, X_i^2, X_iX_j, X_j^2$ and interaction terms $\frac{X_j - X_i}{K_{in}}$, the equations discovered by the LLC and LLC +TPSINDy are closer to the true multi-dimensional governing equations (Fig. 3e), and can achieve smaller predictive errors across all nodes (Fig. 3f). Especially as the prediction time increases (up to $100T$), our LLC can still produce accurate trajectories with periodicity, while the TPSINDy finds it difficult to capture appropriate system behavior (Fig. 3g), demonstrating that discovering the true dynamic equations from data is crucial for improving long-term predictions. Moreover, we test our LLC on a BA network and two empirical networks, namely, the *C. elegans*[59] and Drosophila[60], to analyze whether different topologies have an impact on the results. Supplementary Fig. 10 shows that it can obtain satisfactory equations and predictive trajectories on both synthetic and empirical topologies. More details can be found in Supplementary Section VIII.A.

The predator–prey (PP) is a heterogeneous system[61], where node state represents the individual's position, but nodes are divided into two roles, i.e., a predator ($i = 0$) and many preys ($i > 0$), resulting in three types of pair-wise relationships, i.e., predator–prey, prey–predator, and prey–prey. Here, we consider a fully connected topology structure. By setting masks $M_x$ and $M_a$ to form inputs for nodes with different types, dynamics equations are generated for the predator and prey, respectively. Note that due to the various types of nodes and edges, we set up multiple neural networks based on heterogeneous physical priors to decompose the mixed network dynamics signals, i.e., $\hat{Q}_0^{(self)}$ and $\hat{Q}_{0,j}^{(inter)}$ for predator, and $\hat{Q}_i^{(self)}$, $\hat{Q}_{i,j}^{(inter)}$, as well as $\hat{Q}_{i,0}^{(inter)}$ for prey. When testing, the prey are randomly dispersed while the predator enters from the top right corner as the initial system state. Over time, the predator opens up a path within the prey

swarm, ultimately leading a confused predator ring equilibrium state (Fig. 4b). The predator catches up with the swarm but becomes trapped at its center, where the prey form a stable concentric annulus around it. In this configuration, the repulsive forces exerted by the predator cancel out due to the symmetry of the system. As shown in Fig. 4a, our LLC can discover a more accurate governing equation compared to the TPSINDy. Although TPSINDy produces the equation that matches the form of the true equation, there is a significant discrepancy between its predictive motion trajectories and ground truth, resulting in an unclosed ring with a large radius (Fig. 4c). Note that the TPSINDy requires pre-defined and reasonable elementary functions, including $\frac{X_j - X_i}{|X_j - X_i|^2}$, $(X_j - X_i)$, etc. In contrast, our LLC does not require such strong prior knowledge and the equation discovered by ours indeed captures the same steady-state behavior of the swarm (Fig. 4d).

## Inferring dynamics of chaotic systems

We further uncover the governing equations for three-dimensional chaotic systems on networks, including Lorenz[62] and Rössler dynamics[63]. To examine the impact of the initial sensitivity of chaotic systems on our LLC, we consider three initial value settings: fixing the initial states of all nodes to 0.1, sampling from a standard Gaussian distribution, and sampling from a uniform distribution U(0, 2). Then, we employ a BA network as the topological structure to generate the dynamics data of the Lorenz. We observe that the governing equations derived from the dynamics data, generated using different initial values, are very similar (Fig. 5a). The predictive states of an attractor of the three discovered equations starting from the same initial values are close to the actual system behavior before 1000 time steps, and then the butterfly effect gradually emerges (see Fig. 5b), which means that our tool can accurately forecast a chaotic system for a long time, that is, 1000 times the training time steps.

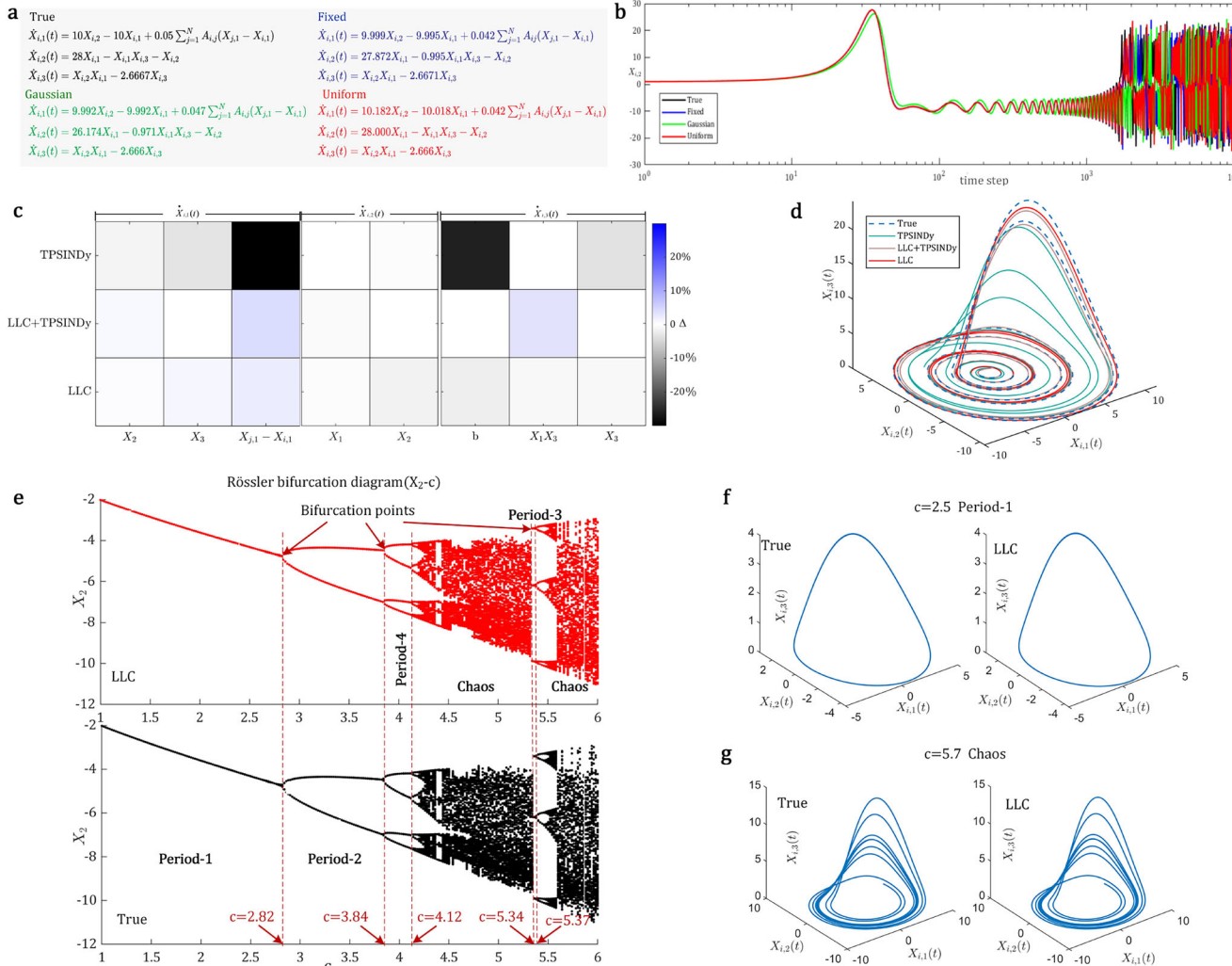

**Fig. 5 | Results of inferring the dynamics of chaotic systems. a** Comparison of governing equations inferred by our LCC under different initial conditions on the coupled Lorenz system. **b** Comparison of predictive states of an attractor with the same initial values, produced by equations inferred by our LLC under different initial conditions. **c** Coefficient errors between the equations inferred by each method on the Rössler system and the true equation. **d** Comparison of states of the same attractor, generated by the governing equations inferred by the TPSINDy,

LCC, and LCC+TPSINDy on the Rössler system. **e** Bifurcation diagram of the Rössler system via the Poincaré section method, with the horizontal axis representing the parameter $c$ (ranging from 1 to 6) and the vertical axis representing the states on the second dimension ($X_{i,2}$) of an attractor. The discovered equation exhibits the same period-doubling and chaotic phenomenon as the true equation. **f** Comparison of limit cycle at period-1, i.e., $c = 2.5$. **g** Comparison of chaos at $c = 5.7$.

For the Rössler system, by comparing with TPSINDy, which is equipped with a solid foundation of function library, including a third-order polynomial ($X_i, X_i^2, X_i^3$) and an interaction term ($X_j - X_i$), our LLC with less prior recovers more accurate governing equations (Fig. 5c) and achieves smaller predictive errors (Fig. 5d). We also analyze the state transition behavior of the Rössler system, and it is evident from the comparison of bifurcation diagrams generated by the inferred and true equations that period-doubling patterns and the bifurcation points are consistent, showing the transition process from period-1 to period-2, to period-4, to chaos, then to period-3, and finally to chaos again (see Fig. 5e). Figure 5f, g also shows the limit cycle at period-1 and chaotic behavior. This confirms the potential of our tool in analyzing the complex behaviors and properties of systems with unknown dynamics. Additional settings, experimental results, and analysis on chaotic systems can be found in Supplementary Section IX.

### Inferring dynamics of empirical systems

As an impressive example of network dynamics, we collect daily global spreading data on COVID-19[64], and use the worldwide airline network

retrieved from OpenFights[65] as a directed, weighted empirical topology to build an empirical system of real-world global epidemic transmission. Only early data, before government interventions (i.e., the first 45 days), are considered here to preserve the intrinsic spread dynamics of the disease. By comparing the number of cases in various countries or regions (Fig. 6a–d), we found that the trend of the results predicted by the TPSINDy is relatively good in countries with many neighbors. However, for countries with fewer neighbors, the results tend to be overestimated in the early stages of disease transmission and underestimated in the later stages. In contrast, our LLC can achieve consistently accurate and robust predictions regardless of the number of neighboring regions, demonstrating the superior comprehensive capability of LLC in handling diverse and heterogeneous network structures across different regions. Additional results for other countries and regions are provided in Supplementary Section X. Based on the recovered equations (Fig. 6e), we see that LLC_each+TPSINDy, LLC_each (where we input observations from each region into the LLC to obtain the equation specific to that region) and LLC_total (where we input observations from all regions into the LLC to derive a common

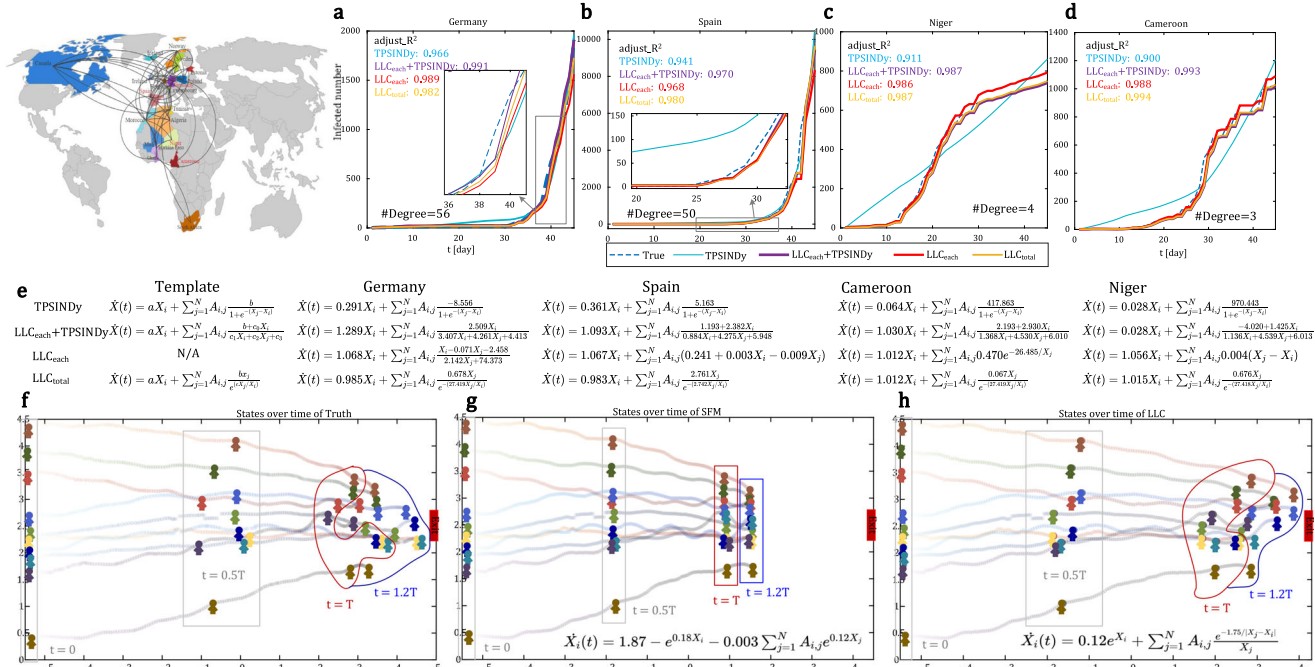

**Fig. 6 | Results of inferring the dynamics of empirical systems, including global COVID-19 transmission and pedestrian dynamics. a–d** Comparison of the number of cases over time in various countries or regions generated by TPSINDy, LLC$_{each}$+TPSINDy, LLC$_{each}$, and LLC$_{total}$. **e** Comparison of governing equations inferred by various methods for four representative countries or regions. Note that the template in the LLC$_{each}$+TPSINDy is the induction of equations generated by our LLC for each node. **f** The real pedestrian movement trajectories over time, where the data before $T$ is used for learning. **g** Inferred results on the pedestrian dynamics using a mainstream social force model (SFM). **h** Inferred results on the pedestrian dynamics using our LLC.

functional form, and then adjust the constants in this common form based on the data from each region) show superior interpretability and predictive accuracy. The fitted curves align closely with the true epidemic trends, and the corresponding adjusted $R^2$ scores are consistently near 1, indicating high-fidelity modeling. Specifically, the discovered dynamics equations emphasize the dominant role of self-dynamics with relatively small interaction terms, which matches intuitive expectations: as the node degree decreases, the influence of neighbors diminishes while the contribution from self-dynamics increases, reflected by a larger coefficient $a$. Moreover, the interaction term we discovered is similar to those found in mutualistic dynamics in ecosystems[17] and aligns with the principles of population evolution, particularly in the equations discovered by LLC$_{each}$+TPSINDy. However, due to the bounded interaction formulation in TPSINDy, where each neighbor's maximum influence is limited to 1, the resulting approximation tends to show a roughly linear increase with the number of neighbors. This can lead to an overestimation of early-stage dynamics and an underestimation later, especially in networks with varying node degrees. Counterintuitively, as the node degree decreases, $a$ decreases while $b$ increases in the TPSINDy, indicating that the growth of cases becomes more influenced by neighbors than by the node itself when the number of neighbors decreases. When predicting the case numbers in Spain by the TPSINDy, $b$ appears to be negative, suggesting neighbors are always impractical negative contributions to oneself. The above results demonstrate that we may have discovered a more suitable transmission model for the global disease.

For the other real-world system, we collect the crowd movement trajectories on a group of people walking in the same direction through the corridor towards a fixed destination[66] and use the k-nearest neighbor method (with $k = 3$) to construct the topological structure the underlying propagation topology. By comparing with mainstream social force models (SFM)[67] in pedestrian dynamics, the equation discovered by ours exhibits a phenomenon close to the ground truth, that is, the crowd gradually moves and gathers towards the destination, experiencing dispersion, aggregation and queuing. In contrast, the learned SFM causes the crowd to move forward side by side, as shown in Fig. 6f–h. Note that the core distinction between the discovered equation and the SFM lies in the inclusion of pedestrian distance in the interaction term, which is in line with the famous territorial effect[68] that pedestrians always hope to maintain a distance from others. This also demonstrates the applicability of ours to real data with unknown pattern of changes.

## Discussion

We introduced a computational tool to efficiently and accurately discover the effective governing ordinary differential equations from observed data of network dynamics. We proposed decoupling network dynamics signals through neural networks and physical priors, and parsing governing equations through a pre-trained symbolic regression, to alleviate the challenges of excessive free variables and high computational cost. Through sufficient experiments on various network dynamics from biology, ecology, epidemiology, genomics, and neuroscience, and by comparing against the latest methods, our LLC can reconstruct the most accurate governing equations with tolerable execution efficiency, even in the face of noisy data and incomplete topologies. It also performs well in complex situations, i.e., multidimensional and heterogeneous systems, such as brain dynamics and predator-prey system, demonstrating its universality and practicality. Interestingly, by analyzing the behavior of chaotic systems and recovering bifurcation diagrams, we demonstrate that it can use the time series from unknown dynamics to gain insight into its properties, such as the existence of multiplication of period. We also show how it can handle real-world datasets, including global epidemic transmission and pedestrian movements, and demonstrate that it can discover more effective symbolic models. In a sense, we view our tool as the

equivalent of a numerical Petri dish to gain insights about unidentified dynamics behind observations, advancing the development for breaking through the inexplicable change mechanisms of complex phenomena.

Although the developed tool provides a feasible solution for the analysis and interpretability of complex systems, how to acquire the underlying topology structures remains an open question. Moreover, many complex systems in real life are not purely autonomous systems, where external disturbances may also change over time[69], exceeding the application boundary of this work. More complex application scenarios are also challenges, such as non-deterministic systems[70], partial differential systems[71], and high-order network interactions[72]. In fact, we have already made a preliminary attempt at higher-order interactions in Supplementary Section VII, and the results indicate that our tool has great potential for handling higher-order interactions, providing strong support for future extensions. Another interesting application of large language models (LLMs) is using them as external tools for LLM-based artificial intelligence agents, assisting in reasoning and decision-making by enabling the efficient discovery of patterns of change from data. Although there is still some way to go in achieving reality-centric artificial intelligence, we believe that developing practical tools, such as the one presented here, will better prepare us for more complex and urgent phenomena such as outbreaks of new infectious diseases and abnormal climate change.

# Methods

## Details of the LLC

In this section, we present the various modules in our LLC, including data acquisition, selection of intervals, differential technology, physical priors, signal decoupling, symbolic regression, and termination conditions.

**Data acquisition.** Generating synthetic data is a crucial step in validating algorithm performance and simulating real-world network behaviors. For a specific dynamics scenario $f$, we determine its hyperparameters and then choose a reasonable topology structure $A$, such as the ER network[51]. After the initial state $X(0)$ is given, we can solve the Initial Value Problem (IVP) using the Runge–Kutta method[73] to simulate the network behavior at any time $t$, i.e., $X(t)$. Note that we solve the IVP over a fixed time interval $t \in [0, T]$ setting both relative error and absolute error thresholds to $10^{-12}$ to ensure high precision throughout calculations. This is a critical requirement in scientific research where accuracy of results holds paramount importance. Please refer to Supplementary Section I for detailed experimental settings of each scenario. Additionally, we collect real-world data for the empirical systems, i.e., global spreading data[64] and the trajectories of pedestrians[66]. More details can be found in Supplementary Section X.

**Selection of intervals.** When inferring the governing equations of complex network dynamics, whether the data has a reasonable sampling range, i.e., $[T', T]$, and the sampling interval, i.e., $\delta_t$, have a significant impact on the inference results. For example, the data in which all nodes reaching a consistent steady state provides little useful information for inferring the governing equation. If the system exhibits periodic behavior, the data we choose needs to cover at least one full period to capture its dynamics. For chaotic systems, a sufficiently long observation period is required to capture the system's long-term behavior and stability, due to its sensitivity to initial conditions. Meanwhile, as different sampling intervals have a slight impact on the results empirically (see Supplementary Section V.B), particularly in scenarios displaying clear non-linearity, such as Kuramoto model, determining the appropriate range and interval for sampling is a practical problem that needs to be addressed. To simplify the optimization process, we set the range to start from 0, i.e., $T' = 0$, and

establish the relationship between $T$ and $\delta_t$ as: $\delta_t = \frac{T}{S}$, where $S$ is a given sampling steps. Here, we employ the simulated annealing algorithm[74] to perform univariate optimization on $T$. The objective function is defined as follows:

$$
T^* = \arg \min_{T > 0} \sum_{i=0}^{N-1} \left[ \sum_{t=0}^{T-\delta_t} \left( \dot{X}_i(t + \delta_t) - \dot{X}_i(t) \right)^2 \right.
$$
$$
\left. + \lambda \sum_{t = T - 10\delta_t}^{T} \left( \dot{X}_i(t) - \dot{X}_i(T) \right)^2 \right], \tag{1}
$$

where $N$ is the number of nodes, and $\lambda$ is an adjustable coefficient. Note that the first term on the right-hand side of the equation controls the avoidance of drastic changes, while the second term ensures that the system reaches a steady state. The objective not only minimizes amplitude fluctuations between adjacent periods but also ensures that changes within the entire period remain as stable as possible.

**Differential technology.** As a core and commonly used preprocessing technique for identifying ordinary differential equations[8,75], numerical differentiation effectively computes approximate derivatives, i.e., $\dot{X}(.)$, from time series of state data, i.e., $X(\cdot)$. The derivatives of measurement variables provide the targets for decoupling network dynamics signals by learning neural networks, i.e., $\dot{X}_i(t) = Q_i^{(self)}(X_i(t)) + \sum_{j=1}^{N} A_{i,j} Q_{i,j}^{(inter)}(X_i(t), X_j(t))$, where the subscripts $i$ and $j$ represent the corresponding nodes, with being $A_{i,j}$ is the adjacency matrix. $Q_i^{(self)}$ and $Q_{i,j}^{(inter)}$ capture the evolution of the system's own state and the dynamics governing pairwise interactions with neighboring nodes, respectively. However, the inadvertent calculation of these derivatives may directly degrade the quality of the learned neural networks. Conventional finite-difference approximations will greatly amplify any noise present in the state data, meaning that even a small amount of noise can lead to a noticeable degradation in the quality of the numerical derivatives[76]. Smoothed finite difference is an effective way to mitigate the effect of noise through smoothing the time series data before calculating the derivative[77], which is also a widely applicable technique that has been empirically validated and recommended for use in identifying ordinary differential equations[8,75,77]. Our empirical research has shown that using a five-point finite difference method, also known as the fourth-order central difference method, yields satisfactory results across various scenarios. This method is particularly advantageous when compared to differential methods of varying accuracies, such as the second-order and sixth-order central difference methods (see Supplementary Section V.C). Therefore, we use a smoothed finite difference method along with a five-point difference approximation to carefully calculate approximate derivatives, minimizing noise as much as possible. Specifically, we use the Savitzky–Golay filter[78] to smooth the original state time series data and then approximate the derivatives through a five point finite difference method as follows:

$$
\dot{X}_i(t) \approx \frac{X_i(t - 2\delta_t) - 8X_i(t - \delta_t) + 8X_i(t + \delta_t) - X_i(t + 2\delta_t)}{12\delta_t}, \tag{2}
$$

where $\delta_t$ is the time interval between data points.

**Physical priors.** A widely recognized physical prior in network dynamics is that the change of network states can be influenced by its own states and the states of its neighbors[6,8,38,39], which can be formalized as

$$
\dot{X}_i(t) = Q_i^{(self)}(X_i(t)) + \sum_{j=1}^{N} A_{i,j} Q_{i,j}^{(inter)}(X_i(t), X_j(t)), \tag{3}
$$

where the subscripts $i$ and $j$ represent the corresponding nodes, $A_{i,j}$ is the adjacency matrix, $\boldsymbol{Q}_i^{(self)}$ and $\boldsymbol{Q}_{i,j}^{(inter)}$ capture the evolution process of their own states and the neighbors' dynamical mechanism governing the pairwise interactions respectively. With the appropriate choice of $\hat{\boldsymbol{Q}}_i^{(self)}$ and $\boldsymbol{Q}_{i,j}^{(inter)}$, Eq. (3) provides a broad description of complex system dynamics[6,40], such as biochemical processes, birth and death processes, heat diffusion, neuronal dynamics[58], gene regulatory[79], as well as chaotic phenomena[3] and oscillations[4] phenomena. For homogeneous dynamics, the self dynamics of all nodes are identical, and those of all pairwise interactions are also the same. That is, we can ignore the subscripts $i$ and $j$, and directly obtain unique values for $\boldsymbol{Q}^{(self)}$ and $\boldsymbol{Q}^{(inter)}$. For heterogeneous dynamics, we can group $\boldsymbol{Q}_i^{(self)}$ and $\boldsymbol{Q}_{i,j}^{(inter)}$ according to heterogeneous topology with $K$ node types and $E$ edge types, as follows:

$$\dot{X}_i(t) = \boldsymbol{Q}_k^{(self)}(X_i(t)) + \sum_{e=1}^{E}\sum_{j=1}^{N} A_{i,j}^e \boldsymbol{Q}_e^{(inter)}(X_i(t), X_j(t)), \qquad (4)$$

where, $1 \le k \le K$ and $A_{i,j}^e$ is an adjacency matrix that only contains edges with type $e$. At this point, we need to learn $K$ self dynamics functions and $E$ interaction dynamics functions. Note that in the extreme case where all nodes have different self dynamics, as long as the above equation satisfies $K = N$.

**Signal decoupling.** To achieve signal decoupling using the physical priors, we parameterize $\boldsymbol{Q}_i^{(self)}$ and $\boldsymbol{Q}_{i,j}^{(inter)}$ using neural networks, i.e., $\hat{\boldsymbol{Q}}_i^{(self)}$ and $\hat{\boldsymbol{Q}}_{i,j}^{(inter)}$. Specifically, we use a multi-layer perceptron to model the nonlinearity of the self dynamics. That is, $\hat{\boldsymbol{Q}}_i^{(self)}(X_i(t)) := \boldsymbol{\psi}^f(X_i(t))$ is a simple feed-forward neural network with $L$ hidden layers, where each hidden layer applies a linear transformation followed by a rational activation function. To capture a wide range of dynamic scenarios, we design a neural network to approximate the interaction dynamics as follows: $\hat{\boldsymbol{Q}}_{i,j}^{(inter)}(X_i(t), X_j(t)) := \boldsymbol{\psi}^{g_0}(X_i(t), X_j(t)) + [\boldsymbol{\psi}^{g_1}(X_i(t)) \times \boldsymbol{\psi}^{g_2}(X_j(t))]$, where $\boldsymbol{\psi}^{g_0}$, $\boldsymbol{\psi}^{g_1}$ and $\boldsymbol{\psi}^{g_2}$ denote non-shared multi-layer perceptrons. The first term on the right-hand side represents a coupled nonlinear interaction term, e.g., $(X_j - X_i) \times X_j$, while the second term represents a decomposable one, e.g., $X_j/(1 + X_i^2)$. In this way, whether the interaction term is decomposable or not, it can be automatically learned. Empirically, such architecture design likely facilitates model learning and optimizes performance, as detailed in Supplementary Section VI. Then, the optimal neural approximation is obtained by minimizing the loss $\mathcal{L}$ between the true derivative $\dot{\boldsymbol{X}}(t)$ and the predicted derivative $\hat{\dot{\boldsymbol{X}}}(t)$, which is calculated using $\boldsymbol{\psi}^f$, $\boldsymbol{\psi}^{g_0}$, $\boldsymbol{\psi}^{g_1}$, and $\boldsymbol{\psi}^{g_2}$:

$$\mathcal{L} = \frac{1}{N \times d}\left[ ||\dot{\boldsymbol{X}}(t) - \hat{\dot{\boldsymbol{X}}}(t)||_1 + \lambda \frac{1}{N-1}\sum_{i=1}^{N}(||\dot{X}_i(t) - \hat{\dot{X}}_i(t)||_1 + \mathbb{E}[\dot{X}_i(t) - \hat{\dot{X}}_i(t)])^2 \right], \qquad (5)$$

where $N$ is the number of nodes, $d$ is the state dimension, and $||\cdot||_1$ is $\ell_1$ norm. The first part of the loss function represents the mean absolute error and the second part represents the variance of the absolute error. The parameter $\lambda$ is a coefficient used for balance. The loss function combines both the mean and the variance, aiming to minimize not only the average difference between predictions and true values but also to control the fluctuation of the error. This approach helps make the model more stable during training, preventing overfitting or underfitting. Note that if the network dynamics are homogeneous, meaning that the governing equations are identical for all nodes, then only two neural networks need to be learned, i.e., $\hat{\boldsymbol{Q}}_{\boldsymbol{\theta}_1}^{(self)} := \hat{\boldsymbol{Q}}_i^{(self)}$ and $\hat{\boldsymbol{Q}}_{\boldsymbol{\theta}_2}^{(inter)} := \hat{\boldsymbol{Q}}_{i,j}^{(inter)}$. To accommodate the heterogeneous dynamics, i.e., Eq. (4), we need to learn multiple neural networks, i.e., $\{\hat{\boldsymbol{Q}}_k^{(self)}\}_{k=1}^{K}$ for

self dynamics and $\{\hat{\boldsymbol{Q}}_e^{(inter)}\}_{e=1}^{E}$ for interaction dynamics. These neural networks have the same architecture but do not share parameters. Detailed architectures and training setups are provided in Supplementary Section II.

**Symbolic regression.** Symbolic regression is a supervised learning method that attempts to discover a hidden mathematical expression $g : \mathbb{R}^m \to \mathbb{R}$ from a series of input-output pairs, i.e., $\{(x_i, y_i), \ldots\}$, where $x_i \in \mathbb{R}^m$ is input feature variables and $y_i \in \mathbb{R}$ is target variable. Transformer-based symbolic regression converts this task into a mapping process, where data sequences are transformed into expression sequences. A data sequence consists of input-output pairs, while an expression sequence represents the corresponding prefix expression[37]. A mathematical expression can be viewed as a tree, where intermediate nodes represent operators and leaf nodes represent variables or constants[80]. Operators can be unary, e.g., sin, cos, tan, exp, or be binary, e.g., $+$, $-$, $\times$, $\backslash$. Then, an expression tree can be converted into a corresponding expression sequence. For example, $\cos(2 \times x_1) + 3 \times x_2$ is formatted as a prefix expression, i.e., $[+, \cos, \times, 2, x_1, \times, 3, x_2]$. By tokenizing the expression sequences and constructing a large dataset of expressions—such as millions of samples—we can pre-train a transformer in an end-to-end manner, thereby integrating a vast amount of equation knowledge. Only a single forward propagation is required when inferring a new equation, significantly reducing the inference time. Due to the unbounded nature of constants in expressions, tokenizing and accurately reconstructing them can be challenging. Therefore, an optional adjustment stage for constants can be added after the expressions are generated by the pre-trained model to improve the accuracy of equation recovery and predictions. Generally, BFGS[44] or stochastic gradient optimization[45] can be used to tune the constants in expressions to ensure inference accuracy. Now, based on the outputs generated by trained neural networks, $\hat{\boldsymbol{Q}}_{\boldsymbol{\theta}_1}^{(self)}$ and $\hat{\boldsymbol{Q}}_{\boldsymbol{\theta}_2}^{(inter)}$, we can use the pre-trained symbolic regression model to parse their corresponding symbolic equations efficiently.

During pre-training, it is typically assumed that the data points are independent of time and sampled independently and identically. However, in our network dynamics scenario, data points are time-dependent. Consequently, directly feeding collected state trajectories into pre-trained symbolic transformers may not yield satisfactory regression results, as it has not been trained on such correlated data points. Therefore, a suitable sampling strategy is required to make the data points as independent as possible. One straightforward strategy is to sample with a uniform time step. However, if the state trajectory contains rapidly changing segments, this approach may fail to capture those behaviors accurately, leading to an uneven distribution of samples. Therefore, to improve the effectiveness of the pre-trained symbolic transformer in identifying complex network dynamics, we employ the K-Means sampling method[81] to select more representative data points. It can better adapt to changes in state trajectories, thereby enhancing sample diversity and ensuring more uniform distribution.

**Termination conditions.** We can see from Fig. 1 that both signal decoupling and the overall discovery process require termination conditions. One termination condition for the signal decoupling process is to train the neural networks until they accurately decouple the self dynamics and interaction dynamics. Typically, this is judged by whether the error on the validation set is below a certain threshold. For the termination condition of the entire discovery process, we check if the difference between the predictive results generated by the identified equation and the actual data is below a specified threshold, indicating that a satisfactory dynamics equation has been found.

## Performance measures

The performance measures for evaluating the methods in this work are as follows:

**Adjusted $R^2$ score.** Adjusted $R^2$ offsets the impact of the number of nodes $N$ on $R^2$, so that the true value range is (0, 1), and the larger the better. The score can be calculated as:

$$\text{adjusted } R^2 = 1 - \frac{(1-R^2)(N-1)}{(N-d-1)},$$

where $R^2 = 1 - \frac{\sum_{i=1}^{N}(X_i(t)-\hat{X}_i(t))^2}{\sum_{i=1}^{N}(X_i(t)-\bar{X}(t))^2}$. $X_i(t)$ and $\hat{X}_i(t)$ are the true and predictive states of node $i$ at time $t$, respectively. $\bar{X}(t)$ is the average of $X_i(t)$ over all node and $N$ is the number of nodes.

**NED.** We adopt the normalized Euclidean distance (NED) to finely evaluate the difference between the predictive states generated by the inferred equation and the true states for each node $i$, defined as follows:

$$\text{NED}(X_i, \hat{X}_i) = \frac{1}{D_{\max}^{0:T}} \sum_{t=0}^{T} \sqrt{\left(X_i(t)-\hat{X}_i(t)\right)^2 + \left(\dot{X}_i(t)-\hat{\dot{X}}_i(t)\right)^2},$$

where $D_{\max}^{0:T}$ represents the maximum Euclidean distance between pairs of true states over the time interval [0, $T$]. $X_i(t)$ and $\hat{X}_i(t)$ are the true and predictive states of node $i$ at time $t$, respectively. $\dot{X}_i(t)$ and $\hat{\dot{X}}_i(t)$ are their respective derivatives.

**Recall.** From the perspective of the symbolic form of equations, we use recall to evaluate the accuracy of the discovered equations, specifically whether a certain function term is present or not. Let $\xi_{\text{true}}$ and $\xi_{\text{pre}}$ denote the true and inferred coefficients under a candidate library of function terms. For example, a library is [1, $x$, $x^2$, $x^3$, $x^4$], If a true equation is $y = 1 + x$, then $\xi_{\text{true}} = [1, 1, 0, 0, 0]$. And if an inferred equation is $y = 1 + x^2$, then $\xi_{\text{pre}} = [1, 0, 1, 0, 0]$. Recall measures the percentage of coefficients successfully identified among the true coefficients and is defined as:

$$Recall = \frac{|\xi_{true} \odot \xi_{pre}|_0}{|\xi_{true}|_0},$$

where $\odot$ represents the element-by-element product of two vectors, $|\cdot|_0$ denotes the number of non-zero elements in the vector.

More performance measures, including MRE (Mean Relative Error), MAE (Mean Absolute Error), L2 error, and Precision, can be found in Supplementary Section IV.

## Data availability

The simulating data generated in this study have been deposited in the database under accession code[82]. Infectious diseases and the real data of pedestrian dynamics through link https://github.com/jpatokal/openflights/tree/master/data and https://ped.fz-juelich.de/database/doku.php access.

## Code availability

The paper is currently under review and the code is available on Code Ocean: https://doi.org/10.24433/CO.3947449.v2.

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

## Acknowledgements

B.Y. is supported by the National Key R&D Program of China (Grant No. 2021ZD0112500), National Natural Science Foundation of China (Grant Nos. U22A2098 and 62172185), and the Major Science and Technology Development Plan of Changchun (Grant No.2024WX05). J.C. is supported by the National Natural Science Foundation of China (Grant No. 62206105), Natural Science Foundation of Jilin Province (Grant No. 20250102211JC), Scientific Research Project of Education Department of Jilin Province (Grant No. JJKH20250118KJ), and Jilin Province Youth Science and Technology Talent Support Project (Grant No. QT202225). The authors are also grateful for the helpful discussion with Xu Xu.

## Author contributions

B.Y. conceived the research; J.C., B.Y. and J.H. designed it; J.H. performed it; J.H. and J.C. analyzed the results, and J.H. and J.C. wrote the manuscript.

## Competing interests

The authors declare no competing interests.
