## [Transparent Peer Review file · Nature Communications]

Learning Interpretable Network Dynamics via Universal Neural Symbolic Regression

Corresponding Author: Dr Jiaxu Cui

Version 0:

Reviewer comments:

Reviewer #1

(Remarks to the Author)

The authors address the problem of learning interpretable models of network dynamics from data. To do this, they combine symbolic regression and deep learning. The authors impose a prior assumption about the structure of the network dynamics, decomposing it into an internal term and a pairwise interaction term, which are approximated from data with deep neural networks. Then the functional form of the two terms is found using symbolic regression with a pre-trained transformer. The effectiveness of the technique is demonstrated with a number of examples, both with synthetic models and empirical data. Some comments:

1. In the results section preamble, the authors introduce M_x and M_a as masks but it is not clear from the text what this means. As far as I can tell, "masks" are not explained anywhere else in the paper. This is confusing.
2. The authors are using a pre-trained transformer for symbolic regression. The authors should address more thoroughly what systems are contained in the training set for the transformer, and whether there is any danger of the systems used to evaluate their method in this manuscript being present in the training set. Many of the systems used in the manuscript are network versions of canonical models, so I suspect there is significant overlap between training and testing sets.
3. In Appendix F Lorenz system is spelled as "Lorzen"
4. The authors should address the limitations of assuming pairwise coupling functions in the network model. This assumes away any multi-node interactions, making a broad class of models impossible to discover with this technique. Network dynamics literature recently points to importance of hypergraphs/multi-node interactions in realistic systems.
5. Many of the examples selected have linear diffusive coupling. Similarly, this should be mentioned explicitly as a limitation of the evaluation.
6. For the pedestrian dynamics dataset, from the description of their method, the network structure should be dynamic, but the method relies on a static network being defined. Can the authors elaborate on this?

Overall, while the presented numerical explorations of the method are comprehensive, the presentation of the method is lacking some clarity for me. Specifically, more care should be given to explaining the choice of transformer and the training data that went into it with more detail, as well as what class of symbolic terms it is constrained to. At the moment this information is discussed in very broad terms, only stating things like "our LLC does not require such strong prior knowledge" and "dataset containing hundreds of millions of equations". Some of the performance comparisons are also puzzling, for example the functional forms in Fig. 6e. It is completely unclear what "masks" are even after reading the paper multiple times. In general I found that the authors explain the TPSINDy method they are comparing against in more detail than their own. Since the authors are utilizing a pre-trained transformer model not created by them, and a reasonably standard decomposition of network dynamics into an internal and a coupling term, the numerical validation of pre-trained transformers for estimation of these terms is in my view the primary contribution of this paper. The performance is very promising, but the authors should address some of the clarity issues.

(Remarks on code availability)

The codebase does not contain a README and mostly runs a scaled down example. A comment within the example instructs the user on how to reproduce the paper results by downloading the pretrained model elsewhere. I did not attempt to run the full code.

Reviewer #2

(Remarks to the Author)

In this paper, the authors proposed a framework based on neural networks to capture network dynamics, and further employed the NSRA symbolic regression to learn the compact governing equations for the system. The proposed approach successfully infers classical homogeneous dynamical equations across multiple domains, including Biochemical, Gene Regulatory, Mutualistic Interaction, Lotka-Volterra, Neural, and Epidemic dynamics. Additionally, it demonstrates potential in inferring heterogeneous dynamical equations and real-world system equations.

I appreciate the authors' extensive benchmarking efforts and the clear visualization of results. Moreover, this work exhibits a notable improvement in inference accuracy. However, my primary concern lies in the novelty of the methodology. The divide-and-conquer strategy has been explored in several prior works, and the symbolic regression component appears to be directly adopted from an existing method. If the authors can convincingly address the following questions, I believe this work could be considered for publication in Nature Communications.

As I mentioned before, my primary concern is the methodological novelty. For instance, Cranmer et al. introduced the divide-and-conquer strategy in their paper "Discovering Symbolic Models from Deep Learning with Inductive Biases" and proposed PySR for learning equation expressions. I suggest that the authors provide a more in-depth analysis and discussion on the advantages of the LLC framework in identifying and separating the self and interaction dynamics. Additionally, I encourage the authors to conduct a comparison between the first part of LLC combined with PySR and a GNN+PySR approach. This would help explore and clarify the key strategies within the proposed framework that contribute to high-accuracy dynamic reconstruction.

This question is closely related to the first one. In the signal decoupling component, I noticed that the coupling simulation in the neural network consists of two parts: $\phi^{g_0}(X_i, X_j)$ and $\phi^{g_1}(X_i) \times \phi^{g_2}(X_j)$. Theoretically, increasing the depth of the first neural network should allow it to approximate the behavior of the second neural network. This raises the question of whether both components are necessary in the current network design. I recommend conducting an ablation study to verify the necessity of this specific network construction. Additionally, I noticed that the first figure mentions neighbor information aggregation. Does this imply that the underlying framework is based on a GNN, or is it a newly designed architecture? This part requires further clarification. At the very least, providing a pseudo-code or a minimal code demo would greatly improve the clarity of the proposed approach.

How does this framework handle observational noise? The authors hypothesize that the robustness to noise benefits from the neural network's fitting ability in signal decomposition. This is an interesting point, especially considering that no explicit denoising mechanisms have been implemented. One possible explanation could be that the loss function used in this work inherently accounts for variance, requiring it to be minimized. If so, this variance minimization might interact with the added zero-mean Gaussian noise over time, effectively canceling out its effect. To better understand the underlying reason, I suggest the authors experiment with different types of noise and analyze the framework's response. This would provide further insights into whether the observed robustness is specific to Gaussian noise or holds more generally across other noise distributions.

The authors employed a highly accurate derivative computation formula, which requires data from 12 time points per evaluation and assumes ideal sampling intervals. While this approach ensures precision in a controlled setting, it does not necessarily reflect real-world scenarios, where data availability is often extremely limited, and data quality cannot be idealized. I would like to see an analysis of how the framework's inference accuracy is affected under different derivative computation accuracies or sampling frequencies. How robust is the model when applied to sparser time-series data, as commonly encountered in practical systems? Such an evaluation would provide valuable insights into the method's applicability beyond idealized conditions.

Minor suggestions:

In real-world epidemic systems, the LLC framework appears to have learned significantly different equations for different countries or regions. This is quite unusual, as epidemic models typically share a common functional form, with variations mainly in parameter values rather than the equation structure itself. Could the authors explore imposing constraints to enforce more consistent equation forms across different regions? Alternatively, if the observed discrepancies are indeed meaningful, a more detailed explanation is needed.

Few presentation issues that should be addressed:

Missing Legends: In Figure 3(b) and 3(c), the absence of legends for blue and yellow dashed lines makes it difficult to interpret the results.

Duplicate Citations: References [70] and [71] appear to be cited redundantly. Please review and correct the citation list.

Code or Pseudo-Code for Clarity: Providing either a code demo or pseudo-code would significantly enhance the accessibility of the proposed framework, making it easier for readers to understand and potentially reproduce the results.

(Remarks on code availability)

Providing either a code demo or pseudo-code would significantly enhance the accessibility of the proposed framework, making it easier for readers to understand and potentially reproduce the results.

Reviewer #3

(Remarks to the Author)

The authors present a computational tool for inferring governing equations in complex network dynamics using neural-symbolic regression. Their approach efficiently discovers governing equations even from noisy or incomplete data, demonstrating applicability across various fields, including physics, epidemiology, and real-world scenarios such as epidemic transmission and pedestrian movement. The paper is well-written and clear, the results appear correct, and the topic is of broad scientific interest. Therefore, the manuscript merits publication.

However, I have the following suggestions to improve the presentation and results:

- * Instead of R^2 , I suggest using adjusted R^2 , which better accounts for model complexity and the number of predictors.
- * The study primarily considers random graphs, but it would be valuable to extend the analysis to scale-free networks, which are prevalent in real-world complex systems.
- * Some equations are complex and could benefit from step-by-step explanations, particularly for readers unfamiliar with symbolic regression.

After addressing these points, I will be happy to re-evaluate the manuscript.

(Remarks on code availability)

Version 1:

Reviewer comments:

Reviewer #1

(Remarks to the Author)

The authors revised the manuscript and thoroughly addressed the comments in my previous review. In particular, I am happy with the addition of details about the mask mechanism which was previously ambiguous. I am also impressed by the authors' thorough response regarding the overlap of training and testing sets. The authors also added experiments including network models with higher order interactions, and clarified how dynamic network structure is handled in the pedestrian example. With these changes the scope and clarity of the manuscript is improved significantly, and I am supportive of publication. There is currently significant interest in equation learning methods and in networked systems, so I believe this manuscript has strong potential for interdisciplinary impact.

(Remarks on code availability)

Reviewer #2

(Remarks to the Author)

I thank the authors for thoroughly addressing my previous comments. The revised manuscript shows substantial improvements, including a significant amount of new computational validation. I greatly appreciate these efforts.

I have only one suggestion for the authors' consideration. The manuscript contains two particularly noteworthy contributions that, in my opinion, deserve greater emphasis in the main text and visualizations, beyond the extensive comparative results currently presented. Specifically:

1. Figure 1 presents the challenge of potential issues in the raw data and proposes a masking mechanism to address these challenges, a novel and valuable insight not discussed in earlier work.
2. The use of simulated annealing to optimize interval selection is also a meaningful methodological contribution with broader implications for the field.

I would also like to thank the authors for providing both pseudocode and executable code. This transparency will undoubtedly help readers better understand the entire pipeline. However, I recommend checking the code to ensure consistency in language, since some comments are currently written in Chinese.

Overall, I believe the manuscript has reached a publishable standard and I support its acceptance.

(Remarks on code availability)

I would also like to thank the authors for providing both pseudocode and executable code. This transparency will undoubtedly help readers better understand the entire pipeline. However, I recommend checking the code to ensure consistency in language, since some comments are currently written in Chinese.

Reviewer #3

(Remarks to the Author)

The authors have improved the paper and solved all the critical points. I therefore recommend publication.

(Remarks on code availability)

Response Letter

Journal Title: Nature Communications

Manuscript ID: NCOMMS-25-06504A

Title of Paper: Learning Interpretable Network Dynamics via Universal Neural Symbolic Regression

We sincerely thank the reviewers and the Editor for their invaluable comments and constructive feedback on improving the quality of the article. We have carefully revised the manuscript based on the suggestions provided, ensuring that all concerns have been thoroughly addressed. We hope the revisions meet your expectations. For clarity, all changes made to the manuscript are highlighted in blue. In this response, we will detail the revisions and provide specific responses to each reviewer's comments. In summary, we have made the following main revisions.

- 1) In response to the Reviewer #1's issue of "Mask", we have provided a detailed explanation and specified the corresponding configuration in the experiments.
- 2) For the concern on the risk of overlap between training set and testing set from the Reviewer #1, we have conducted an analysis and determined that only simple forms in certain scenarios have been included in the training set. In contrast, more complex forms of testing network dynamics have not appeared in the training data at all. This highlights the ability of pre-trained models to predict emerging new equations, given a substantial amount of diverse training data.
- 3) Regarding the spelling issue mentioned by Reviewer #1, we have made corresponding modifications in the main text.
- 4) In line with the reviewer #1's comments of the multi-node interaction, we have added two higher-order interaction experiments. The results empirically confirm that our LLC is still effective in inferring high-order network dynamics.
- 5) In response to reviewer #1's concern on the limitation of the evaluation due to numerous linear coupling terms, we have analyzed the coupling terms across all tested network dynamics scenarios and found that the ratio of linear to nonlinear coupling terms was 4 to 11, indicating that nonlinear coupling represented the majority. Our LLC demonstrated excellent performance in both linear and nonlinear coupling scenarios, highlighting its adaptability.
- 6) Regarding the reviewer #1's concern about clarifying how to handle the dynamic topology of the pedestrian dataset, we clarify that our LLC can indeed handle dynamic topology and have provided corresponding detailed processing steps.
- 7) In response to the reviewer #2's concern on methodological novelty and suggestion of adding experiments to help clarify the key strategies of the proposed framework, we have clarified our contributions and supplemented the experimental results on comparing first part of LLC combined with PySR and a GNN+PySR approach. The results demonstrate the effectiveness of our decoupling module.
- 8) In accordance with the reviewer #2's comments of divide-and-conquer strategy, we have given a detailed explanation on the question about the difference between the decoupling method proposed in this work and the method used in paper entitled "Discovering Symbolic Models from Deep Learning with Inductive Biases".
- 9) Regarding the reviewer #2's comment for clarifying the design of signal decoupling module, we have added detailed ablation experiments as suggested to demonstrate the necessity of each

module.

- 10) Based on the reviewer #2's concerns about the impact of noise types on the results, we have added comparative experiments on various noise types to demonstrate the processing capability of the framework proposed in this work under different experimental settings.
- 11) In accordance with the reviewer #2's suggestions on supplementing differential calculation methods with different accuracies and sampling intervals, we have conducted the experiment to demonstrate the impact of these factors on the results.
- 12) In response to the reviewer #2's suggestion on imposing constraints to enforce more consistent equation forms across different regions in real-world epidemic system, we have incorporated the assumption that the propagation equations of all regions are isomorphic, and presented the experimental results, which further confirmed the effectiveness and flexibility of our LLC.
- 13) Regarding the issues with the missing legends and duplicate citations pointed out by Reviewer #2, we have made revisions in the main text.
- 14) Following the reviewer #2's comments, we have added the pseudo-code implemented in the article in the appendix to enhance the accessibility of the proposed framework and make it easier for readers to understand.
- 15) In response to the reviewer #3's suggestion on replacing R^2 with adjusted R^2 , we have completed the recalculation of all results.
- 16) Regarding the concern on extending to scale-free networks from the reviewer #3, we have clarified that our topologies used in the work includes a number of scale-free networks and have listed specific types of networks for better understanding.
- 17) For the reviewer 3's suggestion to include step-by-step explanations for some equations, we have added a table to enhance readability, offering detailed descriptions for all scenarios used.
- 18) Taking into account all reviewers' comments, we have thoroughly corrected all typographical and grammatical errors throughout the paper. Additionally, we have made further revisions to the main text to address any potential writing issues, ensuring improved readability and coherence.

All our corrections and modifications have been highlighted in the revised manuscript. Below, we provide detailed responses to the reviewers' and the Editor's comments. The reviewers' and Editor's comments are presented in italicized text, while our responses are in regular text.

I. RESPONSE TO REVIEWER 1	3
II. RESPONSE TO REVIEWER 2	11
III. RESPONSE TO REVIEWER 3	23

I. RESPONSE TO REVIEWER 1

Comments to the Author: The authors address the problem of learning interpretable models of network dynamics from data. To do this, they combine symbolic regression and deep learning. The authors impose a prior assumption about the structure of the network dynamics, decomposing it into an internal term and a pairwise interaction term, which are approximated from data with deep neural networks. Then the functional form of the two terms is found using symbolic regression with a pre-trained transformer. The effectiveness of the technique is demonstrated with a number of examples, both with synthetic models and empirical data.

RESPONSE: We sincerely appreciate the time and effort you invested in reviewing our manuscript. We have carefully addressed all of your comments and concerns as outlined below.

Q1: In the results section preamble, the authors introduce M_x and M_a as masks but it is not clear from the text what this means. As far as I can tell, “masks” are not explained anywhere else in the paper. This is confusing.

RESPONSE: Thank you for pointing out our lack of clarity in the description of the masks including M_x and M_a in Section Results. Masks are used in our work to refer to the indicator of missing cases in the observed data, as shown in Fig. 1.

Fig. 1. An illustration for $O = \{X(t), A, M_x, M_a\}$. $X(t) \in \mathbb{R}^{T \times N \times d}$ stores all observed time series and the mask M_x has the same shape as $X(t)$, indicating which nodes has been observed (its values on the corresponding position is set to 1, otherwise it is 0). $A \in \{0,1\}^{N \times N}$ represents the topological structure. M_a and A have the same shape, indicating that the edge is valid during the experiment (its values on the corresponding position is set to 1, otherwise it is 0).

- Role of masks:** The introduction of observation mask M_x and topology mask M_a can effectively distinguish the real observation and missing data, and make full use of the known node topology structure to reduce the impact caused by missing data or incomplete structure. This mechanism can enhance the robustness and generalization ability of the LLC, ensuring that it can still maintain high prediction performance and adaptability in the face of irregular, partially missing or unknown structure data.

- **Specific mask settings** for different scenarios are listed as follows:

Table 1. Mask Settings in Different scenarios

Experiment scenarios	Settings for M_x	Settings for M_a
One-dimensional network dynamics (Section 2.2)	$M_x(t, :, :) = \begin{cases} 0, & t \geq 0.5T \\ 1, & \text{otherwise} \end{cases}$	$M_a(i, j) = 1$
Multi-dimensional network dynamics (Section 2.3)	$M_x(t, :, :) = \begin{cases} 0, & t \geq 0.5T \\ 1, & \text{otherwise} \end{cases}$	$M_a(:, :, t) = 1$
Heterogeneous network dynamics (Section 2.3)	predator: $M_x(:, i, :) = \begin{cases} 1, & i = 0 \\ 0, & \text{otherwise} \end{cases}$ prey: $M_x(:, i, :) = \begin{cases} 1, & i \geq 1 \\ 0, & \text{otherwise} \end{cases}$	predator: $M_a(i, j) = \begin{cases} 1, & i = 0 \wedge j \in i's \text{ neighbours} \\ 0, & \text{otherwise} \end{cases}$ prey: $M_a(i, j) = \begin{cases} 1, & i \geq 1 \wedge j \in i's \text{ neighbours} \\ 0, & \text{otherwise} \end{cases}$
Chaotic systems (Section 2.4)	$M_x(:, :, :) = 1$	$M_a(:, :, :) = 1$
Empirical pedestrians dynamics (Section 2.5)	$M_x(:, :, :) = 1$	$M_a(:, :, :) = 1$
Empirical epidemic spread (Section 2.5)	Total: $M_x(:, :, :) = 1$	Total: $M_a(:, :, :) = 1$
	Each: $M_x(:, i, :) = \begin{cases} 0, & i \neq \text{ID of the region to be inferred} \\ 1, & \text{otherwise} \end{cases}$	Each: $M_a(i, j) = \begin{cases} 0, & i \neq \text{ID of the region to be inferred} \wedge j \in i's \text{ neighbours} \\ 1, & \text{otherwise} \end{cases}$
Evaluation on robustness (Appendix D)	Noise state: $M_x(:, :, :) = 1$	Noise state: $M_a(:, :, :) = 1$
	Topology missing: $M_x(:, :, :) = 1$	Topology missing: $M_a(i, j) = \begin{cases} 0, & i \geq 0.5N \wedge j \in i's \text{ neighbours} \\ 1, & \text{otherwise} \end{cases}$

ACTION: We have added a detailed introduction to the mask settings in each scenario in Appendix A.4 (page 5 in SUPPLEMENTARY INFORMATION). Thank you for pointing out this issue that may have affected understanding, and we believe this change helped improve the overall clarity of the manuscript.

Q2: *The authors are using a pre-trained transformer for symbolic regression. The authors should address more thoroughly what systems are contained in the training set for the transformer, and whether there is any danger of the systems used to evaluate their method in this manuscript being present in the training set. Many of the systems used in the manuscript are network versions of canonical models, so I suspect there is significant overlap between training and testing sets.*

RESPONSE: Thanks for your comments. We fully understand your concern about the possible overlap between the training set and the evaluation systems, which in turn affects the fairness of the results. We have explained this in more detail in the revised manuscript, and added the following clarifications and experimental analysis:

- **What systems are contained in the training set:** We randomly synthesize a large number of expressions as the training set for transformer model. Each randomly generated expression tree contains at most five non-leaf nodes, and its operators and leaf types are sampled from a weighted distribution. After executing the variable dependency rule, we convert the expression from prefix notation to infix notation, simplify it with Python package ‘‘Sympy’’, and insert constant placeholders for later regression. In fact, the specific details are consistent with the process of producing expressions in the paper entitled ‘‘Neural symbolic regression that scales’’.
- **Limited overlap with training set:** After generating the training set, we further verified whether the items of expressions in the experimental network dynamics scenarios in this paper appear in

the training set. We conducted a matching statistics on the overlap between the testing scenarios and the expressions in the training set. The results show that although some basic symbol forms such as $X_j - X_i$, $X_i X_j$ or $\sin(X_j - X_i)$ may exist in the training data, the more complex composite forms did not appear in the training set. In particular, many typical evaluation systems, such as $\exp(|X_k - X_j|)[(X_j - X_i) + (X_k - X_i)]$, $\frac{X_i X_j}{5+0.9X_i+0.1X_j}$, $\frac{1}{1+\exp(-(X_j-1))}$, etc, exhibit complex or multivariate interactions that far exceed the simple symbolic expressions in training set. This also demonstrates that pre-trained models on a large number of equations may emerge to handle complex tasks.

- **Clarification of key contribution:** The key contribution of this work is the development of a novel pipeline including the data acquisition, automatic selection of sampling intervals, finite-difference, signal decoupling, and symbolic regression for automatically discovering network dynamics from data, with a special emphasis on modular decomposition of these dynamics into self and interactive components. Instead of applying a transformer directly to the entire system for symbolic regression, the proposed framework utilizes the transformer only after this decomposition, concentrating on recovering local symbolic expressions. Furthermore, the front module (signal decomposition) in the pipeline significantly influences the back module (symbolic regression). The key innovation here is that by learning to separate self and interaction dynamics, the complexity of symbolic discovery is greatly reduced. This reduction enables even search-based methods, such as genetic programming (GP), to succeed in complex environments. To validate this, we conducted a comparative experiment using LLC+GP and GNN+GP, neither of which employed a symbolic transformer during inference. As indicated in the response to Reviewer #2, Question 1, LLC+GP consistently achieves higher fidelity in recovering the underlying governing equations, especially on more challenging benchmarks. This demonstrates that the advantages arise primarily from the decomposition strategy itself, rather than from any memorization by the pre-trained transformer, which mainly contributes to improving efficiency.

ACTION: We thank you for asking this critical question, which led us to more rigorous validation of our data construction versus our experimental setup, and to refine our explanations. We have recently provided a supplementary explanation of the above in Appendix C (page 7 in SUPPLEMENTARY INFORMATION). We hope that the above additions have effectively cleared your mind about the possibility of train-test overlap.

Q3: In Appendix F Lorenz system is spelled as "Lorzen"

RESPONSE: Thank you for identifying a typo in Appendix F where "Lorenz system" was incorrectly spelled as "Lorzen." We have corrected this error in the revised version and thoroughly proofread the entire text to prevent similar issues.

Q4: The authors should address the limitations of assuming pairwise coupling functions in the network model. This assumes away any multi-node interactions, making a broad class of models impossible to discover with this technique. Network dynamics literature recently points to importance of hypergraphs/multi-node interactions in realistic systems.

RESPONSE: Thank you for pointing out the limitations of the pair-wise coupling assumption in our work. While pairwise coupling functions have been demonstrated to cover a wide range of network dynamics scenarios, the multi-node interactions (e.g., triples, community synchronization) are at the forefront of current research in network dynamics and play an important role in such systems as neural

systems, ecological networks, and social systems. In response to your suggestion, we have added two network dynamics scenarios that include multi-node interaction terms to validate the effectiveness of our LLC.

- **Inferring network dynamics with multi-node interactions:** We have conducted a new series of experiments on higher-order dynamical systems, using systems with real triplet interaction terms as the ground truth. We aimed to verify whether our LLC can accurately identify and retrace these higher-order terms without explicitly restricting the interaction order. The specific high-order systems used are the continuous dynamic development of consensus processes ^[1,2] in higher-order networks, which are described as listed in Table 2. Table 3 lists the equations recovered by our LLC and Fig. 2 shows the results under two scenarios in terms of various evaluation indicators. We see that by properly extending the signal decoupling module to support ternary interaction (adding a new neural network to parameterize the ternary interaction), the higher-order parts can be accurately inferred by our LLC, showing its flexibility, that is, our LLC can be effectively applied to handle high-order network dynamics with multi-node interactions.

Table 2. On the description of higher-order dynamics scenarios.

Scenarios	Governing equations	Description	Parameter setting
Higher-order dynamics 1 ^[1]	$\dot{X}_i(t) = \sum_{j,k} A_{ijk} \exp(l X_k - X_j) [(X_j - X_i) + (X_k - X_i)]$	$x_i(t)$ is the state of node i at time t (such as opinion, preference, etc.), and $A_{ijk} \in \{0,1\}^{N \times N \times N}$ is a 3-hypergraph adjacency tensor representing triadic interactions. $\exp(l X_k - X_j)$ is a nonlinear scaling function that modulates the effects of j and k on node i . l is a parameter that controls the sensitivity of opinion differences; if $l > 0$, the node prefers to synchronize with neighbors with similar opinions (homophily), and if $l < 0$, the node may be influenced by neighbors with large opinion differences (heterophily). $(X_j - X_i) + (X_k - X_i)$ indicates that the state of node i is jointly influenced by j and k , similar to the action of the higher-order Laplacian.	We set $l = 1$. Under the random network topology complement setting, we generate random trajectories for $N = 100$, $T = 0.03$ and $dt = 0.0001$.
Higher-order dynamics 2 ^[2]	$\dot{X}_i(t) = -\mu X_i + \gamma \sum_{j,k} A_{ijk} X_j X_k (1 - X_i)$	$X_i(t) \in [-1,1]$ represents the opinion tendency of an individual i , and $A_{ijk} \in \{0,1\}^{N \times N \times N}$ is a 3-hypergraph adjacency tensor representing triadic interactions. μ denotes the individual opinion decay rate and γ denotes the group coupling strength. When the opinion of individual i is consistent with the group opinion ($X_j X_k > 0$), its opinion is shifted towards the group direction, and the nonlinear term $(1 - X_i)$ prevents the opinion from spilling over the boundary.	We set the parameters as follows: $\mu = 0.3$, $\gamma = 1$, $N = 100$, and the network topology is a random network.

Table 3. Inference results for higher-order dynamics

	Higher-order dynamics 1	Higher-order dynamics 2
Ture	$\dot{X}_i(t) = \sum_{j,k} A_{ijk} \exp(l X_k - X_j) [(X_j - X_i) + (X_k - X_i)]$	$\dot{X}_i(t) = -0.3X_i + \sum_{j,k} A_{ijk} X_j X_k (1 - X_i)$
LLC	$\dot{X}_i(t) = \sum_{j,k} A_{ijk} \exp(X_k - X_j) [(X_j - X_i) + (X_k - X_i)]$	$\dot{X}_i(t) = -0.298X_i + \sum_{j,k} A_{ijk} X_j X_k (1.006 - X_i)$

[1] Battiston F, Cencetti G, Iacopini I, et al. Networks beyond pairwise interactions: Structure and dynamics[J]. Physics reports, 2020, 874: 1-92.

[2] Majhi S, Perc M, Ghosh D. Dynamics on higher-order networks: A review[J]. Journal of the Royal Society Interface, 2022, 19(188): 20220043.

Fig. 2. Performance evaluation of two high-order dynamics scenarios under six evaluation indicators.

ACTION: We have included experiments on high-order network dynamics in Appendix G (page 15 in SUPPLEMENTARY INFORMATION) to verify that our LLC can accurately identify and regress these high-order structures without explicitly limiting the interaction order. We sincerely thank you for this important suggestion, as it not only helped clarify the boundary of our work but also encouraged us to expand the experiments and enhance the flexibility of our approach.

Q5: Many of the examples selected have linear diffusive coupling. Similarly, this should be mentioned explicitly as a limitation of the evaluation.

RESPONSE: Thank you for pointing out the use of linear diffusion coupling in our experiments. We recognize that while this form of coupling is common in certain classical network dynamics, it may not fully illustrate the adaptability of our method to the coupling mechanisms found in more complex systems. We would like to clarify that, although some systems do utilize linear diffusion coupling, we also incorporate a variety of nonlinear coupling structures in our experiments. In fact, **the ratio of nonlinear to linear coupling in our testing systems is 11:4**. The specific description is shown in Table 4. The experimental results in our work indicate that the proposed LLC consistently recovers the network dynamics equations accurately, regardless of whether the system exhibits linear or nonlinear coupling.

Table 4. Description of coupling term types

Coupling term type	Specific classification	Network dynamics	Governing equations
Linear	Linear combination	FHN	$\dot{X}_{i,1}(t) = X_{i,1} - X_{i,1}^3 - X_{i,2} - \epsilon \sum_{j=1}^N A_{i,j} \frac{(X_j - X_i)}{K_{in}}$
		Rosler	$\dot{X}_{i,1}(t) = -X_{i,2} - X_{i,3} + \epsilon \sum_{j=1}^N A_{i,j} (X_j - X_i)$
		Lorenz	$\dot{X}_{i,1}(t) = 10(X_{i,2} - X_{i,1}) + \epsilon \sum_{j=1}^N A_{i,j} (X_j - X_i)$
		Heat	$\dot{X}_i(t) = 0.5 \sum_{j=1}^N A_{i,j} (X_j - X_i)$
Nonlinear	Multiplicative coupling terms	Bio	$\dot{X}_i(t) = 1 - X_i + \sum_{j=1}^N A_{i,j} (X_i * X_j)$
		Gene	$\dot{X}_i(t) = -2.000X_i + \sum_{j=1}^N A_{i,j} \frac{X_j^2}{1 + X_j^2}$

		LV	$\dot{X}_i(t) = 0.5X_i - X_i^2 - \sum_{j=1}^N A_{ij}X_jX_i$
		Epi	$\dot{X}_i(t) = -X_i + \sum_{j=1}^N A_{ij}X_j(1 - X_i)$
		High-order 2	$\dot{X}_i(t) = -0.3X_i + \sum_{jk} A_{ijk}X_jX_k(1 - X_i)$
	Asymmetric/directional coupling	MI	$\dot{X}_i(t) = -0.2X_i^3 + 1.2X_i^2 - X_i + 1 + \sum_{j=1}^N A_{ij} \frac{X_iX_j}{5 + 0.9X_i + 0.1X_j}$
		Predator-prey	$X_0(t) = \frac{c}{N} \sum_{j=1}^N \frac{X_j - X_0}{ X_j - X_0 ^2}$ $\dot{X}_0(t) = b \frac{X_i - X_0}{ X_i - X_0 ^2} + \frac{1}{N} \sum_{j=1}^N \left(\frac{X_j - X_i}{ X_j - X_i ^2} + a(X_j - X_i) \right)$
	Nonlinear excitation coupling	Kuramoto	$\dot{X}_i(t) = w_i + \sum_{j=1}^N A_{ij} \sin(X_j - X_i)$
		Neural	$\dot{X}_i(t) = -X_i + \sum_{j=1}^N A_{ij} \frac{1}{1 + \exp(-(X_j - 1))}$
		High-order 1	$\dot{X}_i(t) = \sum_{jk} A_{ijk} \exp(l X_k - X_j) [(X_j - X_i) + (X_k - X_i)]$
		Real-world datasets	Showing obvious nonlinearity, and the inferred coupling terms contain $\frac{bX_j}{e^{(cX_j/X_i)^r}}$, $\frac{c_0X_i+b}{c_1X_i+c_2X_j+c_3}$, $\frac{aX_i}{X_j+b'}$, $\frac{X_j(aX_i-b)}{X_j^2+c}$ and so on.

ACTION: In the revised manuscript, we have added clarification on this bias in Appendix A.1 (page 2 in SUPPLEMENTARY INFORMATION) and indicated that in the future, we plan to systematically extend our LLC to more diverse network coupling mechanisms, including controllable state dependent coupling and non-local interactions.

Q6: For the pedestrian dynamics dataset, from the description of their method, the network structure should be dynamic, but the method relies on a static network being defined. Can the authors elaborate on this?

RESPONSE: Thank you for your valuable questions, your feedback is very helpful for us to further clarify the applicability of our approach. We are happy to provide further explanations regarding the issue of dynamic network structure in the pedestrian dynamics dataset.

- **Clarify the ability of our LLC to deal with dynamic network structure:** It is a valid observation that the network structure in the pedestrian dynamics dataset is dynamic and changes over time in real situation. Since most scenarios involve a static topology, we simplify the topology structure to a constant $A := A(1) = \dots = A(t)$ for ease of understanding. In fact, our approach can handle both static and dynamic network structures, i.e., the topology A can change over time, represented as $A(t)$.
- **Building dynamic topology structures in the pedestrian dynamics dataset:** For node i , we dynamically *select* K nearest nodes as its neighbors at time t as (Fig. 3 illustrates an example):

$$\mathcal{N}_i(t) = \{j | j \in \text{argtopk}_{j \neq i}^K(-d(X_i(t), X_j(t)))\}, \quad d(X_i(t), X_j(t)) = \|X_i(t) - X_j(t)\|_2$$

Fig. 3. Schematic diagram of dynamic topology for K-nearest neighbor construction network. The value of K is 3.

- **The prior template of the governing equation of network dynamics with dynamic topology:**

$$\dot{X}_i(t) = Q_i^{(self)}(X_i) + \sum_{j=1}^N A_{ij}(t) Q_{ij}^{(inter)}(X_i, X_j), A_{ij}(t) = \begin{cases} 1, & \text{if } j \in \mathcal{N}(i) \\ 0, & \text{otherwise} \end{cases}$$

- **The principle behind dynamic topology processing:** Given that we know $X_i(t)$ at any time t , i.e., the positions of each person, we can dynamically construct $A_{ij}(t)$ based on this positions when inferring the governing equation. Then, by utilizing the signal decoupling module, we can separate the self dynamics and the interaction dynamics using the previously mentioned prior template. Finally, we employ symbol regression to identify the corresponding white-box expressions.

ACTION: In the revised manuscript, we have added detailed instructions in Pedestrian Dynamics in Appendix J.2 (page 24 in SUPPLEMENTARY INFORMATION) that elaborate on how our LLC specifically handles dynamic network structure. We believe these clarifications can answer your question.

Q7: Overall, while the presented numerical explorations of the method are comprehensive, the presentation of the method is lacking some clarity for me. Specifically, more care should be given to explaining the choice of transformer and the training data that went into it with more detail, as well as what class of symbolic terms it is constrained to. At the moment this information is discussed in very broad terms, only stating things like “our LLC does not require such strong prior knowledge” and “dataset containing hundreds of millions of equations”. Some of the performance comparisons are also puzzling, for example the functional forms in Fig. 6e. It is completely unclear what “masks” are even after reading the paper multiple times. In general I found that the authors explain the TPSINDy method they are comparing against in more detail than their own. Since the authors are utilizing a pre-trained transformer model not created by them, and a reasonably standard decomposition of network dynamics into an internal and a coupling term, the numerical validation of pre-trained transformers for estimation of these terms is in my view the primary contribution of this paper. The performance is very promising, but the authors should address some of the clarity issues.

RESPONSE: We sincerely thank the reviewer for their thoughtful and detailed feedback. We address each concern point-by-point below:

Justification for Transformer Choice: We acknowledge the need for a more detailed justification of the use of Transformers. In fact, In Section 2.2 of the manuscript, we have conducted comparative experiments across several Transformer-based symbolic regression approaches. Under the premise that

the first stage of LLC successfully decouples the system dynamics, the results consistently show that the selected NSRA method achieves superior performance in both regression accuracy and computational efficiency compared to alternative methods. Based on this empirical evidence, we chose NSRA as the symbolic regression model in our framework. We have further clarified this in the revised manuscript to improve the overall clarity and completeness of our method description.

Clarification of the Transformer Model and Training Data: We have described in detail the training set generation process and the overlap with the test scenes. The results show that there is no large overlap in the training set. So there is no contradiction between "hundreds of millions of equations" and "strong prior knowledge". Finally, we illustrate that the advantage of our approach does not rely on the powerful training set of Transformer-based methods, but rather on the decoupling capability of the framework.

Clarification on Fig. 6e: In the figure, the first column illustrates four different settings: (1) the epidemic equations identified using TPSINDy; (2) the templates obtained by the LLC framework under human-guided intervention, followed by coefficient estimation using a two-stage optimization approach; (3) the templates generated by treating epidemic data as homogeneous, with coefficients subsequently optimized individually for each country; and (4) the distinct epidemic equations derived by treating the data as heterogeneous from the outset. The second column presents the templates corresponding to each of these approaches, and the subsequent four columns depict the resulting epidemic equations identified for each country.

Clarification of “Masks” and Notation: In Appendix A.4, we have further explained the Mask and illustrate the Settings in different scenarios.

Analysis of experimental results: We appreciate the reviewer’s comment regarding the perceived imbalance in the comparison discussions. In the revision, we have more carefully and fairly compared our method and TPSINDy, ensuring that the description of our own method is more comprehensive and transparent than that of competing baselines.

Clarifying Key Contributions: The main contribution of this paper is : (1) proposes a unified pipeline for network dynamics identification, which integrates tailored data representations, the automatic selection of sampling intervals, finite difference estimation, signal decoupling strategy, and pre-trained symbolic regression model. The framework effectively addresses the challenges posed by high-dimensional network dynamics data and significantly enhances the efficiency and applicability of traditional symbolic regression methods in this context. (2) Leveraging physical priors from the minimal model of network dynamics, we design a decoupling framework that separates the governing equations into self-dynamics and interaction dynamics, thereby enhancing the interpretability and manageability of complex system modeling. (3) We employ a pre-trained transformer-based symbolic regression model to convert time-series data into expression sequences, significantly reducing inference time while enhancing the accuracy of recovering governing equations. (4) This study conducts extensive evaluations across diverse dynamical systems, including those from physics, biochemistry, ecology, and epidemiology, demonstrating that LLC consistently achieves strong performance across a wide range of scientific domains.

II. RESPONSE TO REVIEWER 2

Comments to the Author

In this paper, the authors proposed a framework based on neural networks to capture network dynamics, and further employed the NSRA symbolic regression to learn the compact governing equations for the system. The proposed approach successfully infers classical homogeneous dynamical equations across multiple domains, including Biochemical, Gene Regulatory, Mutualistic Interaction, Lotka-Volterra, Neural, and Epidemic dynamics. Additionally, it demonstrates potential in inferring heterogeneous dynamical equations and real-world system equations.

I appreciate the authors' extensive benchmarking efforts and the clear visualization of results. Moreover, this work exhibits a notable improvement in inference accuracy. However, my primary concern lies in the novelty of the methodology. The divide-and-conquer strategy has been explored in several prior works, and the symbolic regression component appears to be directly adopted from an existing method. If the authors can convincingly address the following questions, I believe this work could be considered for publication in Nature Communications.

RESPONSE: Thank you very much for your positive comments on our work. We greatly value the concerns raised by you regarding the methodological novelty and thank you for your in-depth review of our study.

Novelty: In fact, divide and conquer strategy has been used in many studies as a general solution. However, our approach has some unique innovations in applying this strategy.

- This work presents a universal neural symbolic regression tool designed to effectively and efficiently identify the governing equations of network dynamics, i.e., LLC. It integrates customized data representations, automatic selection of sampling intervals, finite difference estimation, signal decoupling strategy, and pre-trained symbolic regression model. This framework addresses the challenges associated with high-dimensional network dynamics data, expert knowledge and computational costs, significantly improving efficiency while ensuring the accuracy of the recovery equations.
- Leveraging physical priors from the minimal model of network dynamics, we design an effective signal decoupling module that separates the governing equations into self-dynamics and interaction dynamics, thereby enhancing the interpretability and manageability of complex system modeling.
- We employ a pre-trained symbolic regression model to open the black-box self- and interaction-parts, significantly reducing inference time while enhancing the accuracy of recovering governing equations.
- This study conducts extensive evaluations across diverse complex systems, including those from physics, biochemistry, ecology, and epidemiology, demonstrating that our LLC consistently achieves strong performance across a wide range of scientific domains. While we still have progress to make toward achieving reality-centric artificial intelligence, we believe that developing practical tools like the one presented here will better equip us to handle more complex and pressing issues, such as outbreaks of new infectious diseases and abnormal climate change.

Innovation in the application of symbolic regression: We are utilizing a pre-trained symbolic regression model that is specifically designed for symbolic regression tasks, where data points are considered independent. However, in our case, we are working with data generated by differential equations, which have time-dependent correlations among the data points. As a result, directly applying this model may lead to a decrease in effectiveness due to a mismatch in data distribution. To address this issue, we employ the practical and effective K-Means method to sample trajectory subsets. This approach maximizes diversity and minimizes correlation between the sampled subsets, allowing us to more effectively identify the governing equations of network dynamics. To our knowledge, this is also the first attempt to apply pre-trained models for discovering governing equations in network dynamics.

Q1: As I mentioned before, my primary concern is the methodological novelty. For instance, Cranmer et al. introduced the divide-and-conquer strategy in their paper "Discovering Symbolic Models from Deep Learning with Inductive Biases" and proposed PySR for learning equation expressions. I suggest that the authors provide a more in-depth analysis and discussion on the advantages of the LLC framework in identifying and separating the self and interaction dynamics. Additionally, I encourage the authors to conduct a comparison between the first part of LLC combined with PySR and a GNN+PySR approach. This would help explore and clarify the key strategies within the proposed framework that contribute to high-accuracy dynamic reconstruction.

RESPONSE: Thank you very much for your attention to the novelty of our methodology and for mentioning the PySR framework proposed by Cranmer et al. and its application to symbolic regression. We strongly agree with your suggestion to further analyze the advantages of the LLC framework for identifying and separating self and interaction dynamics. In response to your comments, we made the following supplement:

- **Difference between GNN+PySR and ours:** GNN+PySR, as presented in the paper "Discovering Symbolic Models from Deep Learning with Inductive Bias," combines physics-oriented Graph Neural Networks (GNNs) with genetic programming techniques. This approach simulates physical laws through message-passing mechanisms and energy functions, while genetic programming is used to construct transparent (white box) representations of the node and edge update functions involved in message passing. In contrast, our LLC as a general neural symbolic regression tool specifically designed for analyzing network dynamics. The first part of our framework utilizes prior knowledge to introduce a signal decoupling module that separately parameterizes self-dynamics and interaction dynamics. This method effectively learns network dynamics and mitigates the common issue of over-smoothing often encountered in graph neural networks. Additionally, we employ pre-trained symbolic regression to efficiently derive control equations, addressing the challenge of low search efficiency typically seen in genetic programming. While both methodologies adopt a divide-and-conquer strategy, our framework—including data acquisition, automatic selection of sampling intervals, finite difference methods, signal decoupling, and symbolic regression—places a greater emphasis on understanding the changing patterns in network dynamics. It has demonstrated significant improvements in both efficiency and accuracy in empirical scenarios.
- **Motivation for the design of signal decoupling (the first part of LLC):** The signal decoupling module is a crucial component in our process, as it directly influences the effectiveness of subsequent symbol regression. When designing this module, we considered the following prior knowledge: 1) self message passing and neighbor passing can cover a range of network dynamic

scenarios^[3], i.e., $Q_i^{(self)}(X_i) := \psi^f(X_i)$ and $Q_{i,j}^{(inter)}(X_i, X_j) := \psi^{g0}(X_i, X_j)$, where ψ^f and ψ^{g0} are neural networks; and 2) the representation of the minimum models of network dynamics can also form a wide range of scenarios^[4], i.e., $Q_i^{(self)}(X_i)$ and $Q_{i,j}^{(inter)}(X_i, X_j) := \psi^{g1}(X_i)\psi^{g2}(X_j)$, where ψ^{g1} and ψ^{g2} are neural networks. Therefore, we can have a network dynamics decoupling learning model parameterized by multiple neural networks as follows: $\dot{X}_i(t) = \psi^f(X_i) + \sum_{j=1}^N A_{i,j}[\psi^{g0}(X_i, X_j) + \psi^{g1}(X_i)\psi^{g2}(X_j)]$, where $\psi^f(X_i)$ is the neural network-parameterized self dynamics. By combining $\psi^{g0}(X_i, X_j)$ and $\psi^{g1}(X_i)\psi^{g2}(X_j)$, network dynamics can be learned and decoupled efficiently and accurately, regardless of whether the interaction term is multiplicative decomposable or not. Our supplementary results on rationality of design indicate that, based on your suggestion for Q2, our design performs best in several scenarios, as illustrated in the response to your Q2.

- **GNN+GP v.s. LLC+GP:** In response to your suggestion to compare the LLC framework with the first part of LLC + PySR (referred to as GP in our experiments) and GNN + PySR (also referred to as GP in our experiments), we have incorporated additional experiments and discussion in Section 1 of the revised version to enhance the contribution of the proposed decoupling module. Here are the details:

Specific settings: The experiments cover multiple network dynamics scenarios, including Biochemical, mutualistic interaction dynamics, Lotka-Volterra and gene regulatory network dynamics. Each generates about 100 time series under the corresponding initialization conditions. We use adjusted R^2 and Recall as evaluation indicators.

Result analysis: From Fig. 4, we can see that LLC and LLC+GP outperform GNN+GP in terms of both evaluation indicators, demonstrating the contribution and effectiveness of our decoupling module design. Table 5 lists the governing equations inferred by GNN+GP, LLC+GP, and LLC on six scenarios. We can observe that GNN+GP can recover the equations in simple scenarios (e.g., Bio, LV and Epi), but it cannot obtain good results in complex scenarios (e.g., Gene, Neural and MI). However, LLC+GP can recover the dynamic equation skeleton well in any scenario. That is to say, the proposed decoupling module designed by introducing the physical prior is more suitable for the learning task of network dynamics scenario than the general graph neural networks.

Fig. 4. Results of adjusted R^2 and Recall for GNN+GP, LLC+GP, and LLC on six network dynamics.

[3] Barzel, B., Barabási, A.-L.: Universality in network dynamics. *Nature physics* 9(10), 673–681 (2013)

[4] Barzel, B., Liu, Y.-Y., Barabási, A.-L.: Constructing minimal models for complex system dynamics. *Nature communications* 6(1), 7186 (2015)

Table 5. Comparison of governing equations inferred by GNN+GP, LLC+GP, and LLC on six scenarios, including Biochemical (Bio), Gene regulatory (Gene), Mutualistic Interaction (MI), Lotka-Volterra (LV), Neural (Neur), and Epidemic (Epi) dynamics.

	Ture	GNN+GP	LLC+GP	Ours
Bio	$\dot{X}_i(t) = 1 - X_i + \sum_{j=1}^N A_{ij}(X_i * X_j)$	$\dot{X}_i(t) = 1.130(1 - X_i) + \sum_{j=1}^N A_{ij}(X_i * X_j)$	$\dot{X}_i(t) = 0.979 - x_i + \sum_{j=1}^N A_{ij}(X_i * X_j)$	$\dot{X}_i(t) = 1 - x_i + \sum_{j=1}^N A_{ij}(X_i * X_j)$
LV	$\dot{X}_i(t) = 0.5X_i - X_i^2 - \sum_{j=1}^N A_{ij}X_jX_i$	$\dot{X}_i(t) = 0.486X_i - 0.998X_i^2 - \sum_{j=1}^N A_{ij}X_jX_i$	$\dot{X}_i(t) = X_i(0.486 - 0.998X_i) - \sum_{j=1}^N A_{ij}X_jX_i$	$\dot{X}_i(t) = 0.500x_i - 1.002x_i^2 - \sum_{j=1}^N A_{ij}X_jX_i$
Epi	$\dot{X}_i(t) = -X_i + \sum_{j=1}^N A_{ij}X_j(1 - X_i)$	$\dot{X}_i(t) = -1.000X_i + \sum_{j=1}^N A_{ij}X_j(1 - X_i)$	$\dot{X}_i(t) = -1.023X_i + \sum_{j=1}^N A_{ij}1.023X_j(1 - X_i)$	$\dot{X}_i(t) = -1.000X_i + \sum_{j=1}^N A_{ij}X_j(1 - X_i)$
Gene	$\dot{X}_i(t) = -2.000x_i + \sum_{j=1}^N A_{ij} \frac{x_j^2}{1+x_j^2}$	$\dot{X}_i(t) = -1.000X_i + \sum_{j=1}^N A_{ij} \frac{x_j^2}{2+x_j^2}$	$\dot{X}_i(t) = -1.999X_i + \sum_{j=1}^N A_{ij} \frac{x_j^2}{1.000+x_j^2}$	$\dot{X}_i(t) = -2.000X_i + \sum_{j=1}^N A_{ij} \frac{x_j^2}{1+x_j^2}$
Neural	$\dot{X}_i(t) = -X_i + \sum_{j=1}^N A_{ij} \frac{1}{1+\exp(-(X_j-1))}$	$\dot{X}_i(t) = -0.596X_i + \sum_{j=1}^N A_{ij}0.237X_j + 0.262$	$\dot{X}_i(t) = -x_i + \sum_{j=1}^N A_{ij} \frac{\exp(X_j)}{2.710+\exp(X_j)}$	$\dot{X}_i(t) = -X_i + \sum_{j=1}^N A_{ij} \frac{\exp(X_j)}{2.712+\exp(X_j)}$
MI	$\dot{X}_i(t) = -0.2X_i^3 + 1.2X_i^2 - X_i + 1 + \sum_{j=1}^N A_{ij} \frac{X_i X_j}{5+0.9X_i+0.1X_j}$	$\dot{X}_i(t) = 0.722X_i^2 - 0.170X_i^3 + 1 + \sum_{j=1}^N A_{ij}0.192X_jX_i + 0.004X_j$	$\dot{X}_i(t) = (-0.197X_i^2 - 5.213)(X_i - 0.521) + 0.456 + \sum_{j=1}^N A_{ij} \frac{X_i X_j}{4.999+0.899X_i+0.100X_j}$	$\dot{X}_i(t) = X_i^2(1.20 - 0.20 * X_i) - X_i + 0.854 + \sum_{j=1}^N A_{ij} \frac{X_i X_j}{4.86+0.90X_i+0.10X_j}$

ACTION: We have added the comparative experiments of GNN+GP and LLC+GP in Section 2.2 (page 4 in MAIN TEXT) and Appendix D.2 (page 10 in SUPPLEMENTARY INFORMATION) in the new version of the manuscript. We believe that these supplements can more fully demonstrate the innovative and practical advantages of the LLC framework. Thank you for your valuable suggestions, which prompted us to further improve the comparative analysis and deepen the contribution of our work.

Q2: This question is closely related to the first one. In the signal decoupling component, I noticed that the coupling simulation in the neural network consists of two parts: $\phi^{g^0}(X_i, X_j)$ and $\phi^{g^1}(X_i)\phi^{g^2}(X_j)$. Theoretically, increasing the depth of the first neural network should allow it to approximate the behavior of the second neural network. This raises the question of whether both components are necessary in the current network design. I recommend conducting an ablation study to verify the necessity of this specific network construction. Additionally, I noticed that the first figure mentions neighbor information aggregation. Does this imply that the underlying framework is based on a GNN, or is it a newly designed architecture? This part requires further clarification. At the very least, providing a pseudo-code or a minimal code demo would greatly improve the clarity of the proposed approach.

RESPONSE: Thank you for your in-depth review and valuable feedback on our framework, especially regarding the need for the design of decoupling component. We have carried out further analysis based on your suggestions and made the following improvements in the revised version:

- **Verification of the necessity of combination of $\psi^{g^0}(X_i, X_j)$ and $\psi^{g^1}(X_i)\psi^{g^2}(X_j)$:** According to your suggestion, we have constructed an ablation experiment to verify the necessity. The specific settings are as follows: we compare the effects of the five variant methods listed in Table 6 on network dynamics scenarios, including Biochemical (Bio), Gene regulatory (Gene), Mutualistic Interaction (MI), Lotka-Volterra (LV), Neural (Neur), and Epidemic (Epi) dynamics.

Table 6. Ablation experiment setup

Variants	Model settings
g_0	It only keeps the $\psi^{g^0}(X_i, X_j)$ to process the input signal for capturing interactive dynamics.
g_0^{deep}	The same architecture as the $\psi^{g^0}(X_i, X_j)$ was used, but the number of hidden layers was set 5 to verify whether the neural network with higher depth could improve the expression effect of interactive information.

$g_1 * g_2$	It uses only the $\psi^{g^1}(X_i)\psi^{g^2}(X_j)$ part of the original design to capture interactive dynamics.
$g_1^{deep} * g_2$	The $\psi^{g^2}(X_i)$ part is kept unchanged on the original $\psi^{g^1}(X_i)\psi^{g^2}(X_i)$ structure, and the number of hidden layers of $\psi^{g^1}(X_i)$ is set 5 to verify the role of $\psi^{g^1}(X_i)$ depth in capturing complex dynamics.
$g_1 * g_2^{deep}$	The $\psi^{g^1}(X_i)$ part is kept unchanged on the original $\psi^{g^1}(X_i)\psi^{g^2}(X_i)$ structure, and the number of hidden layers of $\psi^{g^2}(X_i)$ is set 5 to verify the role of $\psi^{g^2}(X_i)$ depth in capturing complex dynamics.

Ablation study

Fig. 5. Comparison results of average performance across six network dynamics scenarios from five variants and our LLC

Result analysis: Fig. 5 shows the average performance of five variants and our LLC in terms of adjusted R2 and Recall. We see that the variants with independent use of $\psi^{g^0}(X_i, X_j)$ and $\psi^{g^1}(X_i)\psi^{g^2}(X_j)$ perform poorly, and the independent increase of depth can indeed enhance the performance, but the improvements are still limited. Our LLC considers both multiplicative decomposable and non-decomposable cases by combining $\psi^{g^0}(X_i, X_j)$ and $\psi^{g^1}(X_i)\psi^{g^2}(X_j)$, which may make model learning easier and obtain the optimal performance empirically. We hope these supplementary results can eliminate your doubt.

- **Concern on the underlying framework is based on a GNN:** We would like to further clarify that our framework is not based on any existing GNN architecture. Our framework as a general neural symbolic regression tool specifically designed for analyzing network dynamics. The first part of our framework utilizes prior knowledge to introduce a signal decoupling module that separately parameterizes self-dynamics and interaction dynamics. This can effectively learn network dynamics and mitigate the common issue of over-smoothing often encountered in graph neural networks. While a GNN-like neighbor information aggregation mechanism is used in some operations, our design is customized and specialized for multi-level adaptive information passing, and it focuses on the capture of self dynamics and interaction dynamics.
- **On providing pseudo-code:** Thanks for the advice. To enhance the transparency and reproducibility of our approach, we have included pseudo-code in Appendix K.

ACTION: We have added the ablation experiments in Appendix F (page 15 in SUPPLEMENTARY INFORMATION) and included pseudo-code in Appendix K (page 25 in SUPPLEMENTARY INFORMATION). We believe that these additions will effectively improve the transparency and understandability of this work, and also help further validate the effectiveness and innovation of our framework.

Q3: How does this framework handle observational noise? The authors hypothesize that the robustness to noise benefits from the neural network's fitting ability in signal decomposition. This is an interesting

point, especially considering that no explicit denoising mechanisms have been implemented. One possible explanation could be that the loss function used in this work inherently accounts for variance, requiring it to be minimized. If so, this variance minimization might interact with the added zero-mean Gaussian noise over time, effectively canceling out its effect. To better understand the underlying reason, I suggest the authors experiment with different types of noise and analyze the framework's response. This would provide further insights into whether the observed robustness is specific to Gaussian noise or holds more generally across other noise distributions.

RESPONSE: Thank you for your insightful suggestions on the noise handling in our framework. Your suggestions are very valuable and motivated us to conduct further experimental analysis of the response to the effect of noise on the framework.

- **Noise handling mechanism:** As you pointed out, although we assume that the neural network is able to exhibit some robustness to noise, and this robustness benefits from its ability to fit the signal decomposition, we did not explicitly introduce the denoising mechanism in the original paper. The loss function you mentioned to understand the minimization of variance is indeed one possible explanation for noise robustness. In our framework, Savitzky-Golay filtering is used to smooth time series data, reducing the impact of noise before computing the derivative. In addition, the complex problem is decomposed into independent fitting of self dynamics, interaction dynamics can effectively mitigate noise propagation between modules. For example, the noise of the edge model may be partially cancelled out when the node model is aggregated. So, as you said, the robustness of the framework may be related to the type of noise.
- **Experimental analysis of various noise types:** To better understand how the framework performs under different noise types, we have conducted new experiments in Kuramoto model under three different noise types, including zero-mean Gaussian noise, Poisson noise, and phase noise, and analyze the response of the framework. The specific Introduction mechanisms of different types of noise into observation data are shown in Table 7.

Table 7. Introduction mechanisms of different types of noise into observation data

Noise types	Implement method	Description
Gaussian	$X_i^{noise}(t) = X_i(t) + \epsilon_{gaussian}$ $\epsilon_{gaussian} \sim \mathcal{N}(0, \sigma^2), \sigma^2 = \frac{\text{Var}(X_i(t))}{10^{SNR/10}}$	The noise is uniformly superimposed at all time points and exhibits continuous and symmetric random fluctuations. The noise is uniformly superimposed at all time points and exhibits continuous and symmetric random fluctuations.
Poisson	$X_i^{noise}(t) = X_i(t) + \epsilon_{poission}$ $\epsilon_{poission} = \text{Poission}(\lambda) - \lambda, \lambda = \max\left(\frac{\text{Var}(X_i(t))}{10^{SNR/10}}, 0\right)$	The noise appears as a discrete disturbance and is more significant in low intensity signals. If the original signal itself is a count, Poisson noise will directly reflect statistical fluctuations.
Phase	$X_i^{noise}(t) = (X_i(t) + \epsilon_{phase}) \bmod 2\pi$ $\epsilon_{phase} \sim \mathcal{N}(0, \sigma^2), \sigma^2 = \frac{\text{Var}(X_i(t))}{10^{SNR/10}}$	Noise causes random shifts in phase values, but a phase jump such as a jump from 2π to 0. May introduce discontinuities due to periodic boundary conditions.

Fig. 6. The robustness of different noise types in Kuramoto model. The horizontal axis represents the amount of noise which can be measured by the signal-to-noise ratio (SNR), and the larger its value, the less noise is added.

- **Result analysis:** From Fig. 6, we can see that our LLC is superior to TPSINDy in noise tolerance, regardless of the noise type. In addition, our LLC shows strong robustness in the face of Gaussian noise, indicating that it can effectively "smooth" the observed data, minimize the variance introduced by the noise, and maintain a relatively accurate dynamics fit. For non-Gaussian noise, our LLC still produces stable results with lower levels of noise. However, the performance of both methods declines compared to their performance with Gaussian noise when higher levels of noise (SNR=30 for Poission noise and SNR=40 for phase noise) are introduced, especially in the case of phase noise. This results suggest that the framework may need further optimization when dealing with non-Gaussian noise, which may be left for future work.

ACTION: We have added more discussion on the impact of noise types on the performance in Appendix E.1 (page 11 in SUPPLEMENTARY INFORMATION) of the revised manuscript. Thank you for your suggestions, which will help us to further improve the robustness analysis of the framework.

Q4: The authors employed a highly accurate derivative computation formula, which requires data from 12 time points per evaluation and assumes ideal sampling intervals. While this approach ensures precision in a controlled setting, it does not necessarily reflect real-world scenarios, where data availability is often extremely limited, and data quality cannot be idealized. I would like to see an analysis of how the framework's inference accuracy is affected under different derivative computation accuracies or sampling frequencies. How robust is the model when applied to sparser time-series data, as commonly encountered in practical systems? Such an evaluation would provide valuable insights into the method's applicability beyond idealized conditions.

RESPONSE: Thanks for your comments on our work. You point out that we use a highly accurate method (based on five time points of data and an ideal sampling interval) in the derivative calculation. We recognize that data availability and quality can often be insufficient in real-world scenarios. Therefore, it is essential to assess the applicability and robustness of the framework when faced with sparse or non-ideal data conditions. We have made the following additions to the revised manuscript regarding the impact of differential accuracy and sampling intervals on performance.

- **Analysis of derivative calculation accuracy and inference performance:** We have added a new experiment to compare and analyze the performance of three finite difference methods, as shown in Table 8. Fig. 7 shows the average results using these three difference methods over six network dynamics scenarios under the same experimental conditions. The results indicate that the framework performs well with medium accuracy when using the fourth-order difference approximation. However, while the sixth-order central difference method can effectively reconstruct the skeleton of equation, its prediction accuracy is lower than that of the LLC method. This discrepancy arises because the LLC requires the recovered equation to be sufficiently smooth and the boundary conditions to be extendable. Additionally, the second-order central method exhibits a large error due to its lower accuracy. Therefore, this work empirically selects the fourth-order difference approximation (a five-point approximate difference method) as the derivative calculation method to strike a balance between accuracy and efficiency.

Table 8. Description of different derivative estimation methods

Differential type	Derivative method	Feature description
Second order central difference method	$\dot{X}_i(t) = \frac{X_i(t+\delta_t) - X_i(t-\delta_t)}{2\delta_t}$	The calculation is simple, but the accuracy is low. And it has a relatively low requirement for the smoothness of the function and is suitable for most smooth functions.
Fourth order central difference method (5-point approximate difference)	$\dot{X}_i(t) = \frac{X_i(t-2\delta_t) - 8X_i(t-\delta_t) + 8X_i(t+\delta_t) - X_i(t+2\delta_t)}{12\delta_t}$	The accuracy is significantly higher than that of the second-order method and it is more sensitive to high-frequency noise. It is required that the function be differentiable within a larger neighborhood.
Sixth order central difference method	$\dot{X}_i(t) = \frac{-X_i(t-3\delta_t) + 9X_i(t-2\delta_t) - 45X_i(t-\delta_t) + 45X_i(t+\delta_t) + 9X_i(t+2\delta_t) - X_i(t+3\delta_t)}{60\delta_t}$	It features ultra-high precision and is suitable for extremely smooth functions, but there may be slight noise in the method data.

Fig. 7. Comparison of the average performance of different finite difference methods in six one-dimensional network dynamic scenarios.

- The influence of sampling frequency (sampling intervals) on robustness:** To evaluate the impact of different sampling frequencies on the performance, we have conducted experiments on two scenarios, i.e., heat diffusion and Kuramoto. The sampling interval δt influences the sampling frequency directly. A larger interval results in a lower sampling frequency, leading to sparser data. The specific sampling time interval δt is from $\{0.0001, 0.001, 0.01, 0.1\}$ to explore the performance of LLC under different sampling frequencies. From Fig. 8, we can see that varying sampling intervals can indeed affect the experimental results. If the sampling interval is too large (e.g. 0.1), the results tend to deteriorate noticeably. Furthermore, different scenarios impact the results in distinct ways, as evidenced by the comparison between the orange and blue lines in Fig. 8. As mentioned in the main text, we actually utilized the simulated annealing method to optimize δt for various scenarios, ensuring the reliability and effectiveness of our framework. Empirically, most of the experimental scenarios in the main text choose 0.01 as the time interval to obtain satisfactory performance.

Fig. 8. Performance comparison of different sampling time intervals in the Heat and Kuramoto scenarios.

ACTION: We have included these additional experiments in Appendices E.2 (page 13 in SUPPLEMENTARY INFORMATION) and E.3 (page 14 in SUPPLEMENTARY INFORMATION) to the revised manuscript to provide fuller empirical support for the applicability of the framework in non-ideal conditions and to answer your important questions. Thanks again for your constructive feedback, which helped us improve this work.

Q5: *In real-world epidemic systems, the LLC framework appears to have learned significantly different equations for different countries or regions. This is quite unusual, as epidemic models typically share a common functional form, with variations mainly in parameter values rather than the equation structure itself. Could the authors explore imposing constraints to enforce more consistent equation forms across different regions? Alternatively, if the observed discrepancies are indeed meaningful, a more detailed explanation is needed.*

RESPONSE: Thank you for your careful review of our work. You pointed out that in the process of our framework (LLC) modeling epidemics in different countries or regions, there are obvious differences in the form of equations learned, which is inconsistent with the setting that traditional epidemic models are usually consistent in functional form and only differ in parameters by region. In response to your concern, we have added the comparison method in real-world epidemic system. The specific methods are described as listed in Table 9.

Table 9. Description of methods in real-world epidemic system

Methods	Description
TPSINDy	A two-stage sparse regression approach to obtain infectious disease equation for each country or region
$LLC_{\text{each}} + \text{TPSINDy}$	We feed observation from each country or region into the LLC to derive the equation specific to that region. Next, we manually summarize these equations to create an approximate skeleton (basis function library), that encompasses all the transmission equations. Following this, we utilize TPSINDy to learn the combination coefficients for the basis functions based on the data from each region, resulting in the final equations.
LLC_{each}	We feed observation from each country or region into the LLC to derive the equation specific to that region.
LLC_{total}	We feed observations from all countries or regions into the LLC to derive the common functional form and then learn the constants in the common functional form based on the data from each region, resulting in the final equations.

Result analysis: Fig. 9 shows the performance comparison of different methods in real-world epidemic system and Table 10 lists the governing equations found by different methods. The performance of $LLC_{each}+TPSINDy$ and LLC_{total} is similar and clearly better than that of $TPSINDy$. This indicates that our newly discovered disease transmission equations are more suitable for describing the epidemic spread. It also aligns with the intuition that while the functional form of each region is the same, the parameters may differ. We also notice that in the case of no form restrictions, the differences in equations learned by LLC in different regions can reflect the heterogeneous epidemiological characteristics, data reporting mechanisms, and behavioral response patterns in reality in some cases. As an example, when we compared the results of some developed and developing countries (e.g., Iceland and South Africa), we found that different lag effects and nonlinear response terms are introduced into the transmission dynamics.

Fig. 9. Performance comparison of different methods in real-world epidemic system

Table 10. Comparison of governing equations found by different methods in real-world epidemic system

	TPSINDy	$LLC_{each}+TPSINDy$	LLC_{total}	LLC_{each}
Templates	$\dot{X}(t) = aX_i + b \sum_{j=1}^N A_{ij} \frac{1}{1+e^{-(X_j-X_i)}}$	$\dot{X}(t) = aX_i + \sum_{j=1}^N A_{ij} \frac{c_0 X_i + b}{c_1 X_i + c_2 X_j + c_3}$	$\dot{X}(t) = aX_i + \sum_{j=1}^N A_{ij} \frac{bX_j}{e^{(cX_j/X_i)}}$	N/A
Iceland	$a = 0.042,$ $b = 708.343$	$a = 1, b = -0.224$ $c_0 = 2.600, c_1 = 2.277$ $c_2 = 2.894, c_3 = 5.444$	$a = 1.015,$ $b = 0.021$ $c = -15.682$	$\dot{X}(t) = 1.060x_i + 2.512e^{-(11.999X_j/X_i)} + \sum_{j=1}^N A_{ij} \frac{e^{-X_j(0.680X_i+24.685)}}{X_i(1.886X_j-X_i+1.689)}$
Canada	$a = 0.080,$ $b = 28.534$	$a = 0.980, b = 66.174$ $c_0 = 9.385, c_1 = 3.157$ $c_2 = 4.400, c_3 = 3.955$	$a = 1.015,$ $b = 0.018$ $c = -27.418$	$\dot{X}(t) = 1.070X_i + \sum_{j=1}^N A_{ij}(0.342X_i - 2.268) \frac{e^{-24.733/X_j}}{X_j}$

Algeria	$a = 0.112,$ $b = 123.971$	$a = 1, b = 2.530$ $c_0 = 3.551, c_1 = 1.130$ $c_2 = 4.534, c_3 = 5.846$	$a = 1.037,$ $b = 0.018,$ $c = -16.059$	$\dot{X}(t) = 1.059X_i + \sum_{j=1}^N A_{ij} 0.02e^{\frac{24519-5.011X_i}{X_j}}$
Burkina Faso	$a = 0.011,$ $b = 922.518$	$a = 1.049, b = 4.028$ $c_0 = 4.150, c_1 = 1.261$ $c_2 = 4.514, c_3 = 5.989$	$a = 1.014,$ $b = 0.068,$ $c = -27.418$	$\dot{X}(t) = 1.057X_i + \sum_{j=1}^N A_{ij} e^{\frac{8.625X_j-4755.94}{X_i}}$
Ghana	$a = 0.076,$ $b = 282.872$	$a = 1.079, b = 2.525$ $c_0 = 3.762, c_1 = 1.137$ $c_2 = 4.536, c_3 = 5.966$	$a = 1.019,$ $b = 0.231,$ $c = -30.440$	$\dot{X}(t) = 1.056X_i + \sum_{j=1}^N A_{ij} \frac{X_j(0.366X_i-2.771)}{X_j^2+2370.96}$
Cote d'Ivoire	$a = 0.067,$ $b = 189.751$	$a = 0.949, b = 5.641$ $c_0 = 2.223, c_1 = 4.119$ $c_2 = 2.905, c_3 = 3.377$	$a = 1.014,$ $b = 0.676,$ $c = -27.418$	$\dot{X}(t) = 1.066X_i + 7.99 - 7.99/X_i + \sum_{j=1}^N A_{ij} \frac{(0.995X_i-2.771)}{X_j-1.031X_i+3095.15}$
Niger	$a = 0.028,$ $b = 970.443$	$a = 1.042, b = -4.020$ $c_0 = 1.425, c_1 = 1.136$ $c_2 = 4.539, c_3 = 6.013$	$a = 1.015,$ $b = 0.676,$ $c = -27.418$	$\dot{X}(t) = 1.056X_i + \sum_{j=1}^N A_{ij} 0.004(x_j - x_i)$
Tunisia	$a = 0.049,$ $b = 77.143$	$a = 1.202, b = 29.153$ $c_0 = 29.292, c_1 = 4.914$ $c_2 = 3.667, c_3 = 3.851$	$a = 1.014,$ $b = 0.068,$ $c = -27.419$	$\dot{X}(t) = 1.059X_i + 1 + \sum_{j=1}^N A_{ij} 0.470e^{-32.989/X_j}$
Belgium	$a = 0.237,$ $b = 8.970$	$a = 0.997, b = 17.603$ $c_0 = 5.261, c_1 = 3.195$ $c_2 = 4.579, c_3 = 3.974$	$a = 1.051,$ $b = 0.068,$ $c = -27.419$	$\dot{X}(t) = \frac{X_i(0.051X_i+6.17)}{0.049X_i+4.909} + \sum_{j=1}^N A_{ij} \frac{0.356X_i}{X_j+102}$
Germany	$a = 0.291,$ $b = -8.556$	$a = 1.289, b = 0$ $c_0 = 2.509, c_1 = 3.407$ $c_2 = 4.261, c_3 = 4.413$	$a = 0.985,$ $b = 0.678,$ $c = -27.419$	$\dot{X}(t) = 1.068X_i + \sum_{j=1}^N A_{ij} \frac{X_i-0.071X_j-2.458}{2.142X_j+74.374}$
Estonia	$a = 0.058,$ $b = 358.731$	$a = 1, b = 14.153$ $c_0 = 4.249, c_1 = 2.093$ $c_2 = 2.591, c_3 = 5.362$	$a = 1.014,$ $b = 1.068,$ $c = -27.419$	$\dot{X}(t) = 1.057X_i + \sum_{j=1}^N A_{ij} \frac{2.298X_i-9.732}{2.142X_j+68.991}$
Ireland	$a = 0.099,$ $b = 153.027$	$a = 1.031, b = 9.625$ $c_0 = -4.025, c_1 = 1.144$ $c_2 = 4.566, c_3 = 5.992$	$a = 1.017,$ $b = 1.067,$ $c = -27.419$	$\dot{X}(t) = 1.070X_i + \sum_{j=1}^N A_{ij} \frac{(X_i-5.211)(X_i-1.327)}{0.142X_j(X_i-1.327)+5.211}$
Luxembourg	$a = 0.047,$ $b = 623.419$	$a = 1.093, b = 1.193$ $c_0 = 2.382, c_1 = 0.884$ $c_2 = 4.275, c_3 = 5.948$	$a = 1.017,$ $b = 0.067,$ $c = -27.419$	$\dot{X}(t) = 1.074X_i + \sum_{j=1}^N A_{ij} 0.006X_j$
Norway	$a = 0.042,$ $b = 515.3854$	$a = 1.092, b = 55.859$ $c_0 = 223.949, c_1 = 2.250$ $c_2 = 3.977, c_3 = 4.234$	$a = 1.019,$ $b = 0.067,$ $c = -27.419$	$\dot{X}(t) = 1.071X_i + \sum_{j=1}^N A_{ij} 0.001X_j$
Poland	$a = 0.072,$ $b = 201.713$	$a = 0.972, b = 10.025$ $c_0 = 9.100, c_1 = 3.183$ $c_2 = 4.401, c_3 = 3.998$	$a = 1.018,$ $b = 0.068,$ $c = -27.419$	$\dot{X}(t) = 1.068X_i + \sum_{j=1}^N A_{ij} 0.00$
Sweden	$a = 0.276,$ $b = 7.410$	$a = 1.008, b = 5.169$ $c_0 = 3.654, c_1 = 1.168$ $c_2 = 4.567, c_3 = 5.985$	$a = 1.085,$ $b = 0.067,$ $c = -27.419$	$\dot{X}(t) = 1.064X_i + \sum_{j=1}^N A_{ij} \frac{0.354+e^{-(0.354X_j+15.327)}}{X_i}$
South Africa	$a = 0.045,$	$a = 0.993, b = 0.129$	$a = 1.017,$	$\dot{X}(t) = 1.073X_i + \sum_{j=1}^N A_{ij} \left(\frac{0.304(X_i-1)}{X_j} - \frac{0.023}{X_j^2} \right)$

	$b = 652.840$	$c_0 = 0.609, c_1 = 3.526$ $c_2 = 4.743, c_3 = 3.975$	$b = 0.067,$ $c = -27.419$	
Cameroon	$a = 0.064,$ $b = 417.863$	$a = 1.030, b = 2.193$ $c_0 = 2.930, c_1 = 1.368$ $c_2 = 4.530, c_3 = 6.010$	$a = 1.012,$ $b = 0.067,$ $c = -27.419$	$X(t) = 1.057X_i + \sum_{j=1}^N A_{ij} 0.470e^{-26.485/X_j}$
Mali	$a = 0.052,$ $b = 201.836$	$a = 0.997, b = 11.505$ $c_0 = 5.874, c_1 = 3.008$ $c_2 = 4.366, c_3 = 3.936$	$a = 0.985,$ $b = 0.676,$ $c = -27.419$	$X(t) = 1.062X_i + \sum_{j=1}^N A_{ij} \frac{-0.359X_i}{0.173X_i - X_j}$
Spain	$a = 0.361,$ $b = 5.163$	$a = 1.093, b = 1.193$ $c_0 = 2.382, c_1 = 0.884$ $c_2 = 4.275, c_3 = 5.948$	$a = 0.983,$ $b = 2.761,$ $c = -2.742$	$X(t) = 1.067X_i + \sum_{j=1}^N A_{ij} (0.241 + 0.003X_i - 0.009X_j)$

ACTION: We have added the new experimental results in Section 2.5 (page 7 in MAIN TEXT) and Appendix J.1 (page 21 in SUPPLEMENTARY INFORMATION) to further systematically investigate how to strike a balance between functional form and flexibility, which can retain the expressive power of data-driven models while more closely combining the general form of traditional theoretical models. Thank you again for asking this critical and insightful question, which helped us to more fully reflect on and extend the application boundaries and plausibility of the LLC framework in real-world systems.

Q6: *Few presentation issues that should be addressed: Missing Legends: In Figure 3(b) and 3(c), the absence of legends for blue and yellow dashed lines makes it difficult to interpret the results. Duplicate Citations: References [70] and [71] appear to be cited redundantly. Please review and correct the citation list.*

RESPONSE: Thank you for carefully pointing out the details in our manuscript.

- **About lack of legend:** In the revised version, we have enhanced the legend of Figure 3 to clearly indicate the meaning of each curve, thereby improving the readability and information integrity.
- **About reference repeated references:** We have thoroughly checked the citations, and have removed duplicates and renumbered it in the revised version to ensure that the citations are accurate and consistent.

Q7: *Providing either a code demo or pseudo-code would significantly enhance the accessibility of the proposed framework, making it easier for readers to understand and potentially reproduce the results.*

RESPONSE: We appreciate the reviewer's suggestion to include pseudo-code or a code demonstration to improve the accessibility and reproducibility of our proposed framework. In response, we have added detailed pseudo-code in the appendix (see Appendix K, page 25 in SUPPLEMENTARY INFORMATION) that outlines the core structure of our dynamics decomposition module. This addition clearly illustrates the roles of the self-dynamics and interaction-dynamics components. Furthermore, to enhance reproducibility, we have made the full implementation code available in the supplementary material and plan to release it publicly upon acceptance. We believe these steps will make our framework more transparent and easier for the community to adopt and build upon.

Algorithm 1 Pseudo-code of LLC

Input:

$X \in \mathbb{R}^{T \times N \times d}$: States of the dynamic system;
 $A \in \{0,1\}^{N \times N}$: Topological structure of network dynamics;
 $M_x \in \{0,1\}^{T \times N \times d}$: Mask for observed states
 $M_a \in \{0,1\}^{N \times N}$: Mask for topological structure

Output:

$Q_i^{(self)}$: symbolic expression for self dynamics
 $Q_{ij}^{(inter)}$: symbolic expression for interaction dynamics

Data Preprocessing:

$\delta t \leftarrow \text{IntervalSelect}(X)$
% Identify the optimal time interval through an iterative process that incorporates simulated annealing to achieve accurate signal decoupling
 $X, \tilde{X} \leftarrow \text{FiniteDifferences}(X, \delta t)$ %five point finite difference as Eq.(2)
 $\text{sparse_A} \leftarrow \text{Sparse}(A)$ %The sparse representation of the topological structure A

Signal decoupling:

$\hat{Q}_i^{(self)} = \psi^f(X_i(t))$ % Fitting self dynamics
 $\hat{Q}_{ij}^{(inter)} = \psi^{\theta_0}(X_i(t), X_j(t)) + \psi^{\theta_1}(X_i(t))\psi^{\theta_2}(X_j(t))$ % Fitting interaction dynamics
 $\Sigma Q_{ij}^{inter} \leftarrow \text{scatter_sum}(Q_{ij}^{inter}, \mathcal{N}(i), \text{dim} = 1, \text{dim_size} = N)$
%Use *scatter_sum* to aggregate neighbor nodes, $\mathcal{N}(i)$ denotes the neighbors of node i
 $\tilde{X} \leftarrow \tilde{X} \odot M_x$ % \tilde{X} is masked before training
 $A \leftarrow A \odot M_a$ %A is masked before training
Train $\psi^f, \psi^{\theta_0}, \psi^{\theta_1}, \psi^{\theta_2}$ with Loss function Eq.(5)

Symbolic Regression:

Input: $\hat{Q}^{(self)}$: Well-fitted self dynamic; $\hat{Q}^{(inter)}$: Well-fitted interaction dynamics;
 $(X_{sub}, \hat{Q}_{sub}^{self}) \leftarrow \text{KMeansSubsample}(X, \hat{Q}^{self})$ %KMeans sampling is performed on self dynamics
 $Q_i^{(self)} \leftarrow \text{SymbolicRegression}(X_{sub}, \hat{Q}_{sub}^{self})$ %Self dynamics expressions effectively generate through a pre-trained symbolic regression method
 $(\xi_{ij_sub}, \hat{Q}_{sub}^{inter}) \leftarrow \text{KMeansSubsample}(\xi_{ij}, \hat{Q}^{inter})$ %KMeans sampling is performed on interaction dynamics, ξ_{ij} denotes (X_i, X_j)
 $Q_{ij}^{(inter)} \leftarrow \text{SymbolicRegression}(\xi_{ij_sub}, \hat{Q}_{sub}^{inter})$ %Interaction dynamics expressions effectively generate through a pre-trained symbolic regression
Return $Q_i^{(self)}, Q_{ij}^{(inter)}$ % Assess whether additional data is required to support the equation, if so, restart **Data Preprocessing**; otherwise, return

III. RESPONSE TO REVIEWER 3

Comments to the Author

The authors present a computational tool for inferring governing equations in complex network dynamics using neural-symbolic regression. Their approach efficiently discovers governing equations even from noisy or incomplete data, demonstrating applicability across various fields, including physics, epidemiology, and real-world scenarios such as epidemic transmission and pedestrian movement. The paper is well-written and clear, the results appear correct, and the topic is of broad scientific interest. Therefore, the manuscript merits publication. However, I have the following suggestions to improve the presentation and results:

RESPONSE: We would like to express our gratitude for the positive feedback and thoughtful comments provided by the reviewer. We have carefully addressed all the points you raised and have

incorporated your suggestions to further refine and strengthen our work. Your feedback has been instrumental in improving the quality and clarity of our research.

Q1: Instead of R^2 , I suggest using adjusted R^2 , which better accounts for model complexity and the number of predictors.

RESPONSE: Thanks for your valuable suggestions regarding evaluation metrics. We completely agree that adjusted R^2 is a better measure of model complexity and the number of predictors. Adjusted R^2 improves upon the traditional R^2 metric by providing a more accurate assessment of model fit while taking into account model complexity and the number of predictors. Based on your feedback, we have updated the evaluation metric to adjusted R^2 in the revised version and have also updated all the experimental results accordingly. The conclusions drawn from the updated results are consistent with those from the original findings.

Q2: The study primarily considers random graphs, but it would be valuable to extend the analysis to scale-free networks, which are prevalent in real-world complex systems.

RESPONSE: You recommend extending the analysis to include scale-free networks in order to better simulate the network characteristics of complex real-world systems. In fact, we have incorporated scale-free networks into various use cases. The specific network topology settings for each scenario are detailed in Table 11 below.

Table 11. Description of network topology structures used in different experimental scenarios

Scenario	Topology
Bio	Erdős-Rényi random graph, Poisson distribution
Gene	Erdős-Rényi random graph, Poisson distribution
MI	Erdős-Rényi random graph, Poisson distribution
LV	Erdős-Rényi random graph, Poisson distribution
Neural	Erdős-Rényi random graph, Poisson distribution
Epi	Erdős-Rényi random graph, Poisson distribution
Multi-dimensional Dynamics (FHN)	Barabasi-Albert, Power-law distribution, Scale-free
	Drosophila, Power-law distribution, Scale-free
	C.elegans, Power-law distribution, Scale-free
Heterogeneous network dynamics (Predator-prey)	Fully connected
Chaotic systems dynamics (Rossler)	Barabasi-Albert, Power-law distribution, Scale-free
Chaotic systems dynamics(Lorenz)	Barabasi-Albert, Power-law distribution, Scale-free
Dynamics of pedestrians	Dynamic topology
Dynamics of COVID19	Real topology, Scale-free
Robustness valuation experiment	Barabasi-Albert, Power-law distribution, Scale-free

ACTION: To clarify this, we have added the table in Appendix A.2 (page 3 in SUPPLEMENTARY INFORMATION).

Q3: Some equations are complex and could benefit from step-by-step explanations, particularly for readers unfamiliar with symbolic regression.

RESPONSE: Thank you for taking the time to review our work thoroughly. We completely agree with your suggestion that gradually explaining the meaning of complex equations is crucial for readers to accurately understand the interpretability of the framework's output. As a result, we have added detailed notes for each scenario in the revised version, as shown in Table 12 (see Appendix A.1, page 2 in SUPPLEMENTARY INFORMATION). We believe these additions will enhance the readability of the work, particularly for readers from various disciplines. We appreciate your valuable suggestions!

Table 12. Illustration of the equation notation for the experimental scenario

Scenarios	Governing equations	Explanation
Bio	$\dot{X}_i(t) = F - BX_i + \sum_j^N A_{ij}(X_i * X_j)$	Biochemical processes within living cells are mediated by protein-protein interactions in which proteins bind to form protein complexes. $X_i(t)$ is the concentration of protein i at time t . A_{ij} is the effective rate constant of interaction between proteins i and j , and F and B denote the average influx rate and degradation rate of proteins i , respectively.
Gene	$\dot{X}_i(t) = -BX_i^f + \sum_{j=1}^N A_{ij} \frac{X_j^h}{1+X_j^h}$	The dynamics of gene regulatory networks can be described by the Michaelis-Menten equation. The node state $X_i(t)$ is the expression level of gene i . Parameter B controls the decay rate. When $f = 1$, the first term on the right-hand side represents degradation, where the expression level of gene i decreases over time. The second term captures genetic activation, where $h \geq 0$ represents the Hill coefficient, quantifying the saturation rate affected by neighboring nodes.
MI	$\dot{X}_i(t) = b + X_i(1 - \frac{X_i}{k}(\frac{X_i}{c} - 1) + \sum_{j=1}^N A_{ij} \frac{X_i X_j}{d+eX_i+hX_j}$	This equation describes the dynamics of interactions between species in ecology. The abundance $X_i(t)$ of a captured species in a mutualistic differential equation system consists of an afferent migration term b , a logical increase in population capacity k , an Allee effect with a cold starting threshold c , and a mutualistic interaction term with the interaction network A .
LV	$\dot{X}_i(t) = X_i(\alpha - \theta X_i) - \sum_{j=1}^N A_{ij} X_j X_i$	The Lotka-Volterra model (LV) describes the population dynamics of species in competition. The node state $X_i(t)$ represents the population size of species i , the growth parameters of species i denoted by α , θ , are both positive constants.
Neural	$\dot{X}_i(t) = -X_i + \sum_{j=1}^N A_{ij} \frac{1}{1+\exp(-\tau(X_j-\mu))}$	This equation describes the firing rate of the neuron. $X_i(t)$ represents the activity level of neuron i , while the parameters τ and μ determine the slope and threshold of the neural activation function, respectively
Epi	$\dot{X}_i(t) = -\delta X_i + \sum_{j=1}^N A_{ij} X_j (1 - X_i)$	Epidemic dynamics can be used to describe the outbreak of infectious diseases. Each node X_i can represent an individual, where the node state $X_i \in [0,1]$ corresponds to the infection probability of node i . The parameter δ represents the rate at which individuals recover from infection.
Kuramoto	$\dot{X}_i(t) = \omega_i + \epsilon \sum_{j=1}^N A_{ij} \sin(X_j - X_i)$	Kuramoto model is a classical mathematical model to study the synchronization phenomenon of coupled oscillators. Where X_i denotes the phase (Angle over time) of the i -th oscillator. Let ω_i denote the natural frequency of the i -th oscillator, which follows a Gaussian distribution. ϵ represents the coupling strength, which controls the strength of the interaction between the oscillators.
FHN	$X_{i,1}(t) = X_{i,1} - X_{i,1}^3 - X_{i,2} - \epsilon \sum_{j=1}^N A_{ij} \frac{(X_j - X_i)}{K_{in}}$ $\frac{dx_{i,2}(t)}{dt} = a + bx_{i,1} + cx_{i,2}$	The FitzHugh-Nagumo model (FHN) is a neuron model that describes the excitatory behavior of neurons. As a simplified version of the Hodgkin-Huxley model, it is primarily used to study neuron action potentials. The first component $X_{i,1}$ denotes the membrane potential containing the self and interaction dynamics, K_{in} is the in-degree of neuron i (denoting the number of afferent connections to node i), and $\epsilon = 0.05$. The

		second component $X_{i,2}$ denotes the recovery variable.
PP	$\dot{X}_0(t) = \frac{c}{N} \sum_{j=1}^N \frac{X_j - X_0}{ X_j - X_0 ^2}$ $\dot{X}_i(t) = b \frac{X_i - X_0}{ X_i - X_0 ^2} + \frac{1}{N} \sum_{j=1}^N \left(\frac{X_j - X_i}{ X_j - X_i ^2} + a(X_j - X_i) \right)$	The predator-prey (PP) model is a heterogeneous system, where the node state represents the position of each individual. Nodes are classified into two roles: a single predator ($i = 0$) and multiple preys ($i > 0$), leading to three types of pairwise interactions: predator-prey, prey-predator, and prey-prey. The interactions between prey are modeled to exhibit paired short-range repulsion and long-range attraction.
Rosssler	$\dot{X}_{i,1}(t) = -X_{i,1} - X_{i,3} + \epsilon \sum_{j=1}^N A_{ij} (X_{j,1} - X_{i,1})$ $\dot{X}_{i,2}(t) = X_{i,1} + aX_{i,2}$ $\dot{X}_{i,3}(t) = b + X_{i,3}(X_{i,1} - c)$	Rosssler system is a set of ordinary differential equations describing chaotic dynamics. The first dimension shows that the change of $X_{i,1}$ is driven by the negative coupling of $X_{i,2}$ and $X_{i,3}$, which is similar to the damping effect. The second dimension shows the variation of $X_{i,2}$ consisting of the linear actuation of $X_{i,1}$ and its own feedback with the coefficient a controlling the strength of the feedback. The third dimension is the key nonlinear term. The constant b provides the base growth rate for $X_{i,3}$, while $X_{i,3}(X_{i,1} - c)$ introduces a threshold mechanism: when $X_{i,1} > c$, $X_{i,3}$ grows exponentially; And vice versa. This "switch" behavior leads to the stretching and folding of phase space trajectories, which is the source of chaos.
Lorenz	$\dot{X}_{i,1}(t) = a(X_{i,2} - X_{i,1}) + \epsilon \sum_{j=1}^N A_{ij} (X_{j,1} - X_{i,1})$ $\dot{X}_{i,2}(t) = X_{i,1}(r - X_{i,3}) - X_{i,2}$ $\dot{X}_{i,3}(t) = X_{i,1}X_{i,2} - bX_{i,3}$	The first dimension indicates that the change in $X_{i,1}$ is driven by the difference between $X_{i,2}$ and $X_{i,3}$, the coefficient a (Prandtl number) controls the transfer rate of the difference, and the second dimension contains two key roles: $X_{i,1}(r - X_{i,3})$ is the nonlinear coupling term that introduces the feedback from $X_{i,3}$ to $X_{i,1}$ into the dynamics of $X_{i,1}$ - $X_{i,2}$ is the damping term, which suppresses the growth of $X_{i,2}$. The parameter r (Rayleigh number) determines the stability of the system. The third dimension $X_{i,1}X_{i,2} - bX_{i,3}$ is the combination of the energy conservation term and the nonlinearity. $X_{i,1}X_{i,2}$ denotes the synergy between $X_{i,1}$ and $X_{i,2}$ to drive growth at $X_{i,3}$.

Response Letter

Journal Title: Nature Communications

Manuscript ID: NCOMMS-25-06504B

Title of Paper: Learning Interpretable Network Dynamics via Universal Neural Symbolic Regression

We sincerely thank the Editor again for the careful consideration of our manuscript “Learning Interpretable Network Dynamics via Universal Neural Symbolic Regression”. We have addressed the comment of Reviewer #2 in the revised Figure 1 and also edited our manuscript to comply with the policies and formatting requirements of Nature Communications as described in the attached Author Checklist.

RESPONSE TO Reviewer #1 :

Remarks to the Author: *The authors revised the manuscript and thoroughly addressed the comments in my previous review. In particular, I am happy with the addition of details about the mask mechanism which was previously ambiguous. I am also impressed by the authors' thorough response regarding the overlap of training and testing sets. The authors also added experiments including network models with higher order interactions, and clarified how dynamic network structure is handled in the pedestrian example. With these changes the scope and clarity of the manuscript is improved significantly, and I am supportive of publication. There is currently significant interest in equation learning methods and in networked systems, so I believe this manuscript has strong potential for interdisciplinary impact.*

RESPONSE: We sincerely thank the reviewer for the positive and encouraging feedback. We are glad that the revisions have addressed the concerns raised, particularly regarding the mask mechanism, training/testing overlap, higher-order network models, and dynamic structures in the pedestrian example. We greatly appreciate your support and recognition of the manuscript's interdisciplinary potential.

RESPONSE TO Reviewer #2:

Remarks to the Author: *I thank the authors for thoroughly addressing my previous comments. The revised manuscript shows substantial improvements, including a significant amount of new computational validation. I greatly appreciate these efforts.*

RESPONSE: We sincerely thank the reviewer for the positive evaluation and thoughtful suggestions. We are glad that the improvements and additional computational validation in the revised manuscript have been well received. We appreciate your careful review and support.

Q1: *I have only one suggestion for the authors' consideration. The manuscript contains two particularly noteworthy contributions that, in my opinion, deserve greater emphasis in the main text and visualizations, beyond the extensive comparative results currently presented. Specifically:*

1. Figure 1 presents the challenge of potential issues in the raw data and proposes a masking mechanism to address these challenges, a novel and valuable insight not discussed in earlier work.

2. *The use of simulated annealing to optimize interval selection is also a meaningful methodological contribution with broader implications for the field.*

RESPONSE: We thank the reviewer for highlighting the importance of the masking mechanism and the interval selection via simulated annealing. In response to your valuable suggestion, we have updated Fig. 1 to more clearly indicate the presence of masked (Unacquired observations) data, by explicitly adding a “Masked” label in the relevant panel. Additionally, we have revised the visualization of the interval optimization step to better highlight the role of simulated annealing. We hope these changes help emphasize the methodological contributions more effectively in the main text and figures.

Q2: I would also like to thank the authors for providing both pseudo code and executable code. This transparency will undoubtedly help readers better understand the entire pipeline. However, I recommend checking the code to ensure consistency in language, since some comments are currently written in Chinese. Overall, I believe the manuscript has reached a publishable standard and I support its acceptance.

RESPONSE: We thank the reviewer for the positive evaluation and kind support for the acceptance of our manuscript. We also appreciate the suggestion regarding code consistency. In response, we have carefully reviewed the released code and revised all comments to ensure they are written in English for clarity and consistency. We hope this will further improve the usability and accessibility of the code for the broader research community.

Remarks on code availability:*I would also like to thank the authors for providing both pseudo code and executable code. This transparency will undoubtedly help readers better understand the entire pipeline. However, I recommend checking the code to ensure consistency in language, since some comments are currently written in Chinese.*

RESPONSE: We thank the reviewer for the positive evaluation and kind support for the acceptance of our manuscript. We also appreciate the suggestion regarding code consistency. In response, we have carefully reviewed the released code and revised all comments to ensure they are written in English for clarity and consistency. We hope this will further improve the usability and accessibility of the code for the broader research community.

RESPONSE TO Reviewer #3 :

Remarks to the Author: *The authors have improved the paper and solved all the critical points. I therefore recommend publication.*

RESPONSE: We sincerely thank the reviewer for the positive feedback and recommendation for publication. We greatly appreciate your thoughtful comments throughout the review process, which have helped us improve the quality and clarity of the manuscript.